behaviour

body tracking, social interaction, orienting, proxemics, interpersonal synchrony

**Author for correspondence:**
Juha M. Lahnakoski
e-mail: j.lahnakoski@fz-juelich.de

†These authors contributed equally to the current study.

# Unobtrusive tracking of interpersonal orienting and distance predicts the subjective quality of social interactions

Juha M. Lahnakoski[1,2,3,†], Paul A.G. Forbes[4,†],
Cade McCall[5] and Leonhard Schilbach[1,6]

[1]Independent Max Planck Research Group for Social Neuroscience, Max Planck Institute of Psychiatry, Kraepelinstr. 2-10, 80804 Munich, Germany
[2]Institute of Neuroscience and Medicine, Brain & Behaviour (INM-7), Research Center Jülich, Wilhelm-Johnen-Straße, 52428 Jülich, Germany
[3]Institute of Systems Neuroscience, Medical Faculty, Heinrich Heine University Düsseldorf, Moorenstr. 5, 40225 Düsseldorf, Germany
[4]Social, Cognitive and Affective Neuroscience Unit, Department of Cognition, Emotion, and Methods in Psychology, Faculty of Psychology, University of Vienna, Liebiggasse 5, 1010 Vienna, Austria
[5]Department of Psychology, University of York, Heslington, York YO10 5DD, UK
[6]Department of Psychiatry and Psychotherapy, Ludwig-Maximilians-Universität, Nussbaumstr. 7, 80336 Munich, Germany

JML, 0000-0002-5223-7822

Interpersonal coordination of behaviour is essential for smooth social interactions. Measures of interpersonal behaviour, however, often rely on subjective evaluations, invasive measurement techniques or gross measures of motion. Here, we constructed an unobtrusive motion tracking system that enables detailed analysis of behaviour at the individual and interpersonal levels, which we validated using wearable sensors. We evaluate dyadic measures of joint orienting and distancing, synchrony and gaze behaviours to summarize data collected during natural conversation and joint action tasks. Our results demonstrate that patterns of proxemic behaviours, rather than more widely used measures of interpersonal synchrony, best predicted the subjective quality of the interactions. Increased distance between participants predicted lower enjoyment, while increased joint orienting towards each other during cooperation correlated with increased effort reported by the participants. Importantly, the interpersonal distance was most informative of the quality of interaction when task demands and experimental control were minimal. These results suggest that interpersonal measures of behaviour

gathered during minimally constrained social interactions are particularly sensitive for the subjective quality of social interactions and may be useful for interaction-based phenotyping for further studies.

## 1. Introduction

Non-verbal behaviour is an essential aspect of social interactions, conveying the mental states of interactants and affecting the quality of social relations. For example, increases in behavioural synchrony, more direct orientation towards an interaction partner and reductions in interpersonal distance all have been linked to a range of prosocial outcomes, including greater cooperation [1,2], affiliation [3], rapport [4] and positive affect [5,6]. Conversely, abnormalities of non-verbal behaviour have also been reported in a range of psychiatric disorders, including schizophrenia [7,8], social anxiety disorder [9] and autism [10–13], which may explain some of the social difficulties faced by persons affected by these conditions.

Interpersonal synchrony has been studied using a range of measures, from behaviour to physiological signals to brain activity (for a brief introduction, see [14]). However, while many studies have shown effects of interpersonal synchrony and the related concept of mimicry, these phenomena may not be observable in all conditions [15,16]. Dynamic coupling of individuals may arise without direct synchrony of behaviour [17] and strict synchrony of movements may even be detrimental to the attainment of shared goals [18]. This suggests that additional mechanisms may be relevant, especially in naturalistic situations during which interaction partners may take different roles and mutually coordinate their behaviour in complementary ways.

Proxemic measures [19], such as distancing and orienting behaviours, could be more sensitive than interpersonal synchrony at reflecting positive (and negative) attitudes towards other people in some social situations. They might also help to quantify more complex and complementary actions that occur in a dyad that may not manifest as synchronous movements. For example, experiments using virtual reality have indicated that distance and orienting behaviour of participants towards ethnic outgroup members is predictive of individuals' implicit prejudice [20] and subsequent aggression towards those avatars [21]. The behaviour of an interaction partner during an interaction may in turn be reflected in reciprocal proxemic behaviour. For example, in a virtual environment, participants may avoid and orient away from agents who had treated them unfairly in a prior interaction [22]. Moreover, attentive versus inattentive orienting towards a partner during a stressful experience is associated with closer physical distance to the attentive partner and higher subjective evaluations of comfort after the experience [6].

The advent of motion-tracking technology has made it possible to precisely quantify how people's movements vary across different social contexts [23,24]. Yet, when capturing interpersonal behaviour, traditional motion tracking technologies, such as electromagnetic motion trackers or inertial measurements units (IMUs), have several limitations [25]. They can be expensive, cumbersome, and may not be appropriate for clinical populations [26]. Markerless optical motion tracking has the benefit that it can be performed from a distance, providing a minimally intrusive measurement environment. Some studies have compared the performance of optical motion trackers, such as the Microsoft Kinect, to 'gold standard' motion trackers and have generally yielded encouraging findings (for a comprehensive comparison, e.g. [27]). For example, Müller *et al.* [28] recently compared the performance of the Kinect v. 2 to a multi-camera Vicon (Oxford, UK) system in gait tracking. They found excellent agreement with the Vicon on a range of gait parameters (e.g. step length, step width and step time) and argued that even subtle changes in gait, such as those associated with preclinical cerebellar ataxia, may be detectable by the Kinect v. 2 sensors. However, in another study comparing Kinect and Vicon systems, a customized set-up employing retro-reflective markers instead of standard skeleton tracking was suggested to be necessary for clinically relevant accuracy of kinematic evaluation [29]. Other studies have looked specifically at the spatial accuracy of Kinect systems. One study evaluating the accuracy of depth measurements with the first-generation Kinect system found an accuracy from a few millimetres up to a few centimetres towards the maximum range of the system compared with a gold standard laser range finder [30]. However, relatively few studies have evaluated the performance of the Kinect or other unobtrusive tracking systems for capturing interpersonal behavioural synchrony or proxemic behaviour during natural dyadic social interactions.

In the current study, we built a motion tracking system based on two inexpensive commercial optical motion sensors to quantitatively assess dyadic social interactions (Kinect v. 2; figure 1*a*). To validate this system, we used wearable IMUs on the head and the dominant hand of the participants to track their

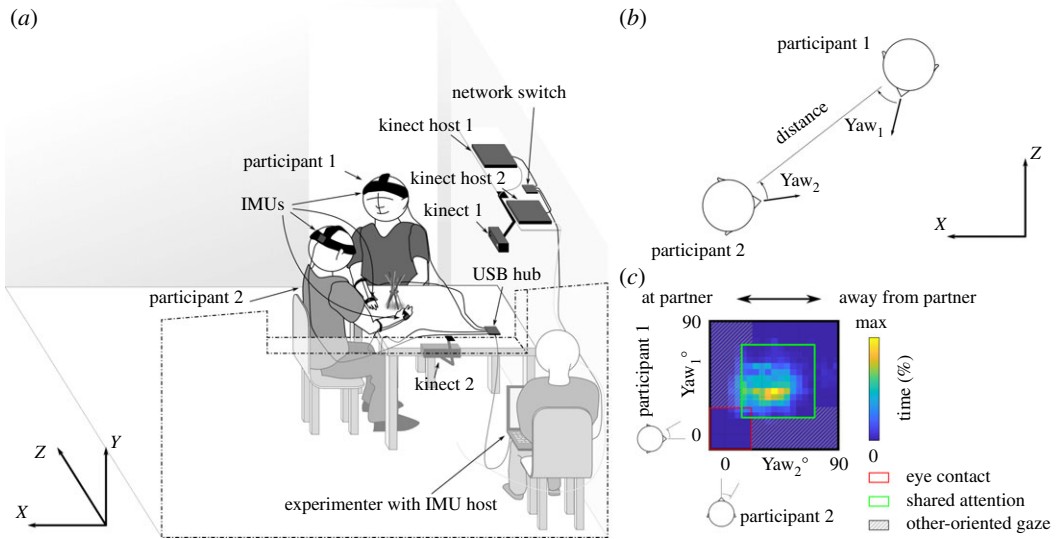

**Figure 1.** Experimental set-up and proxemic analysis. (*a*) Participants sat diagonally on adjacent sides of a table with IMU sensors attached to their head and dominant hand. Two Kinect sensors tracked their movements from a distance. Sensors were attached to three host computers that were connected through a local area network. Experimenter sat in the corner of the room, started the tracking on Kinect host 1 using a wireless numeric keyboard and used the IMU host to insert timestamps at the beginning and the end of each trial. (*b*) We calculated the yaw angle (rotation around the vertical axis) for each participant in relation to the line connecting the heads. Thus, yaw angle of 0 would correspond to the participant looking at their partner. (*c*) We used the yaw angles to build a two-dimensional relational gaze histogram [22] where different parts of the plane are expected to correspond to different gaze behaviour (e.g. eye contact near the origin, shared attention around the centre of the plot, and other-oriented gaze near the edges).

orientation, angular velocity and acceleration. Our goals were twofold: (i) to verify that we can apply the new, unobtrusive system to study proxemic behaviour across two interacting persons, and (ii) to identify behavioural correlates of subjective evaluations of natural interactions and how these are affected by task type or increasing familiarity over the experiment. To study behaviour during natural social interaction, we applied the system to track the body and head movements of participants sitting at a table engaged in two types of social interactions: (i) conversation on two given topics, which were separated by (ii) playing the game 'Keep it steady', where participants pick out wooden sticks from an unstable bundle played alternately in *cooperative* and *competitive* conditions. The discussion topics (see Material and methods) were chosen because they were used in previous studies using video-based motion analysis [5,31] giving us a point of comparison. The game was chosen because it encourages interaction and requires large-scale coordinated movements that should be possible to track using the motion tracking system. The gameplay conditions were carefully matched for overt behaviour, such as turn taking, to minimize trivial differences between gameplay conditions.

After the task, participants evaluated the social interaction experience with the subject impression questionnaire from the intrinsic motivation inventory [32], which is designed to assess participants' subjective experience during laboratory experiments using five subscales: relatedness, enjoyment, perceived choice, pressure and effort. Additionally, they filled out two forms measuring their social traits (autism quotien (AQ) [33] and anticipatory and consummatory interpersonal pleasure scale (ACIPS) [34]) using a web-based questionnaire system.

We estimated interpersonal synchrony of movement as well as dyadic measures of proxemic behaviour, such as joint orienting behaviour [22] and interpersonal distance, to produce a highly compressed, yet readily interpretable summary of complex dyadic behaviour over several minutes of interaction (figure 1*b,c*; for a video demonstration of the proxemic and synchrony measures, see electronic supplementary material, video S1). Additionally, the direction of gaze of the participants (towards the partner, the target or neither) during the tasks was manually annotated from video frames extracted at random moments of each interaction type by two human observers to identify four classes of gaze behaviours (*eye contact, joint attention, other-oriented gaze* and looking *away*; see Material and methods). The annotated gaze labels were used to train a classifier to identify these gaze behaviours based on head location and orientation data without the need for eye-tracking equipment.

We expected interpersonal rather than individual measures of behaviour to be particularly revealing of the quality of the interactions and social traits of the participants, because prosocial outcomes have been associated with interpersonal synchrony [35] and proxemic behaviours [6]. We hypothesized that the level of prosocial behaviours, particularly interpersonal synchrony and eye contact, would be increased from the first conversation in the beginning of the experiment to the second conversation task after participants were more familiar with each other. We also expected there to be reliable differences in prosocial behaviours between *competitive* and *cooperative* gameplay despite careful matching of overt behavioural requirements. Furthermore, we investigated the extent to which the level of cooperation and experimental control through the semi-structured gameplay tasks affected interpersonal coordination of behaviour by comparing the measures of behaviour during different types of task trials and free conversation. The gaze and orienting behaviour between gameplay and conversation tasks were expected to be trivially different due to different task demands. However, we expected the level of cooperation to also be differentially reflected in interpersonal synchrony and gaze behaviour, potentially showing increased synchrony and joint gaze with increasing level of cooperation. In addition, we expected to see different behavioural correlates of the participants' subjective evaluations of the interactions depending on (i) the task demands due to the gameplay tasks, (ii) increased familiarity over the duration of the session, and (iii) level of cooperation during the gameplay.

# 2. Material and methods

## 2.1. Participants

We recruited 20 healthy German-speaking same-sex dyads for the study, but one dyad was lost due to schedule conflicts and data for one dyad was not saved correctly due to technical problems, leaving 18 dyads (13 female, 5 male; 1 left-handed dyad; mean age $26.0 \pm 5.6$, range 20–50, age data for four people was incomplete) for the analyses.

## 2.2. Procedure

During the experiment, participants were first instructed to discuss (for 3–5 min) one of two topics adapted from previously published studies: 'Plan a holiday that should visit at least three countries including how to get there' [31] or 'Plan a three-course dinner that neither of you likes' [5]. Because we were interested in the difference of motion measures between the first and second conversation (and not the topic of conversation), we counterbalanced the order of the topics across dyads.

After the initial discussion task, the participants played six rounds of 'Keep it steady' that involves the participants removing coloured sticks from an unstable bundle one at a time. The game was played alternately in two conditions. In the *cooperative* condition, participants were instructed to take turns to select a stick that the *other* would pull out and try to keep the bundle standing as long as possible. In the *competitive* condition, the participants were again instructed to select a stick that the other player would pull out, but this time they were to make the other player topple the bundle with as few moves as possible while keeping it upright themselves. Because we were interested in (non-trivial) differences between the two gameplay conditions, we carefully matched the required movements (turn-taking, pointing, stick removal).

After the first discussion task of the experiment, before seeing the instructions for the *cooperative* and *competitive* tasks, the subjects were allowed to familiarize themselves with the game for one round with no specific task (*practice* condition). During the gameplay and conversation tasks, the participants were instructed to avoid asking the experimenter questions. Between the different tasks, there was a short pause (mean duration across dyads ranging from 50 s to 130 s; single dyad durations minimum 32 s, maximum 260 s) during which the experimenter gave instructions for the next task and/or reset the bundle of sticks on the table. To include a condition without any explicit task demands, we included data during the pauses, when participants sat upright in a neutral position, as an exploratory analysis of interindividual distance; however, these data were not used for analysis of synchrony or orienting data, because during the pauses the participants also looked at and talked with the experimenter.

## 2.3. Motion tracking

The motion of the participants was tracked with two systems; one based on two Kinect v. 2 sensors with Kinect for Windows adapters (Microsoft, Redmond, WA, USA) and one based on four inertial

measurement units (IMUs; IMU Brick 2.0, Tinkerforge, Stukenbrock, Germany) attached to the head and the dominant hand of the participants with straps designed for action cameras (figure 1a). The two tracking systems were controlled by three laptop host computers: a separate host for each Kinect and one host controlling the IMUs, which was also used to manually insert timestamps to the data at the beginning and end of each trial. The host computers formed a closed local area network through a gigabit Ethernet switch with no external network connections. The recording was initiated by one of the Kinect controllers (*Master* host) that sent a user datagram protocol (UDP) packet to the broadcast address of the network. The packet contained a timestamp of the starting time of recording on the *Master* computer and a filename indicating where the data log should be saved. The other hosts then replied with their own timestamps. The round trip time was measured to be less than 100 ms for all but one dyad (2870 ms). Because of the dyad with the extended delay in the data transfer, we temporally realigned the data of all dyads based on the main peak in the velocity cross-correlation function between the data from the two Kinects for the participant that could be tracked by both sensors. The hosts were running custom software programmed in C#, which supports synchronization of as many Kinect hosts as the network subspace allows. Thus, it is also possible to increase the number of participants that can be reliably tracked by increasing the number of trackers.

## 2.4. Preprocessing

Due to the different sampling rates of the system and intermittent dropped data from the Kinects, the data were downsampled and interpolated to a 10 Hz resolution (from mean sampling rate of $17.0 \pm 5.5$ Hz for the Kinects, and $62.5 \pm 0.0$ Hz for the IMUs[1]). Data were further low-pass filtered with a Gaussian kernel (sigma = 0.5 s) to reduce high-frequency noise. The data from the two Kinect sensors were registered to a common coordinate system based on the locations of four joint locations (head, neck, right shoulder and spine at the shoulder) of the participant that was visible in both Kinects in each dyad (participant 2). Because the view from one of the trackers was blocked briefly by the experimenter between each round of the experiment when he walked to reset the game, the IDs of the participants were randomly assigned by the Kinect software after each occlusion. Therefore, we relabelled the participants using a support vector machine (SVM) classifier with a radial basis function kernel trained on manually labelled data from one of the dyads. For finding the rigid transformation between coordinate spaces, we included only those time windows that contained data from both Kinect sensors. The translation vectors and rotation matrices were calculated based on the mean displacement of the data points between sensors and the singular value decomposition of their covariance matrix following a standard procedure for landmark-based registration [36]. Starting and ending times of the trials for each dyad were checked manually by loading the corresponding video frames saved by the Kinects and adjusted manually to correct for mistakes in the original timestamps (e.g. double key presses by the experimenters or missing markers).

## 2.5. Behavioural variables

After the experiment, participants rated their interaction on a subject impression questionnaire of the intrinsic motivation inventory [32]. Additionally, they filled out web-based versions of questionnaires measuring their personality including the AQ [33] and ACIPS [34]. Two participants did not finish the questionnaires and were excluded from the analyses correlating the scores with behaviour but were included in other analyses.

## 2.6. Data analysis

Data were analysed in Matlab R2017a (MathWorks, Inc., Natick, MA, USA). To compare the conditions, condition-specific mean effects were first calculated for each dyad. Where applicable, the overall effects of the conditions were first evaluated using repeated measures ANOVAs. For the pairwise comparisons between conditions, we employed paired *t*-tests. Because some of the data and residuals were not normally distributed, we repeated the analyses with randomly permuted data. Using this surrogate data as a null distribution yielded results almost identical to the parametric test. Therefore, in the following, we focus only on the parametric test statistics.

---

[1]Due to the 1 ms resolution of the log file, the calculated sampling rate is 62.5 Hz, although the true sampling rate was set to 60 Hz.

### 2.6.1. Interpersonal face and body orientation

For measuring the interpersonal face and body orienting behaviour, we built two-dimensional histogram heatmaps of the face (or body) angle of one interactant as a function of the other (figure 1*b,c*). The orientations were calculated in a moving coordinate system in relation to the line connecting the heads of the participants at each moment in time. Thus, the origin corresponds to the situation where both people are orienting their face directly at the other partner and increasing angles on the *x* (or *y*) axis indicate the participant looking further away from their partner. The value at each pixel of the heatmap corresponds to the amount of time (% of total) the participants were jointly orienting to that combination of directions.[2] The direction of the upper body was estimated as the normal at the middle of the line connecting the shoulders of the participants. The normal vector was constrained to horizontal plane and oriented to the side of the participant's chest. The analyses for the body and head orientations were otherwise identical.

### 2.6.2. Validation of face orientation and gaze behaviour

For validation of the face orientation data and relating it to instances of specific gaze behaviours, such as *eye contact* and *joint attention*, we loaded 50 data points and associated video frames from each condition (*conversation*, *cooperative* and *competitive*; data were sampled randomly from all trials in each condition) in each dyad and two independent raters annotated, with two binary questions, whether each of the participants was looking at their partner or a potential target of joint attention (generally, the joint target was the bundle of sticks in the gameplay conditions, but we also included the instruction cards or, in the case of one dyad, an imaginary object both participants were looking and gesturing at as shared targets). This yielded a total of four binary labels for 2700 samples of joint face directions. For those time points, where the two raters disagreed (see results for the agreement rate between raters), a third rater was employed to break the tie. The third rater rated each binary question separately (look at the partner/target) and the majority (two-thirds) vote for each question was selected. We used the face orientation data at the sampled time points to build two-dimensional distributions of joint face orientations corresponding to (i) *eye contact* (both participants looking at their partner), (ii) *joint attention* (both looking at the target), (iii) *other-oriented gaze* (one person looking at the other while other looks at the common target), or (iv) one or both looking *away* (for prevalence of all combinations of gaze behaviours, see electronic supplementary material, table S1). These data were further extended by additionally extracting the head pitch and location as well as the estimated gaze vector (unit vector in the direction calculated using the yaw–pitch–roll triad). Combinations of these data points (yaw, yaw + pitch, yaw + pitch + location, yaw + pitch + location + gaze vector) were used to train naive Bayesian classifiers to detect the four gaze categories using random balanced subsamples of the data. We compared the performance of these features in the electronic supplementary material, results and focus only on the latter, best performing set of features in the results. The goal of the classifier was to test whether we can use a small subset (few seconds) of the data to make predictions about the full data (tens of minutes). To evaluate this, we performed 50 cross-validation iterations using half of the combined samples from all participants for training the classifier and the other half for testing. The classifier was then also applied to the full data to predict the overall amount of gaze behaviours in the full data. Additionally, we tested the generalizability of the classifier similarly, but this time using leave-one-dyad-out cross-validation, training on all-but-one dyad and testing on the left-out dyad. This process was repeated for each dyad and average performance across dyads and categories was calculated. We report the average class accuracies rather than overall accuracy, because the categories could not be balanced for the individual dyads inflating the overall accuracy due to high accuracies for the overrepresented categories.

### 2.6.3. Proxemic behaviour

Interpersonal distance was calculated as the distance between the heads of the participants in each dyad at each time point. The mean distance values for the distance are calculated over all time points in each

---

[2]Because the difference in face angle when orienting toward the partner or the shared target changes drastically as the participants move (particularly when moving closer or further away from the target or rotating around it to see it from another side), we also normalized the angles by the difference between the direction of the target and the partner to validate our findings. However, due to the lack of a landmark for the target, the normalized results were noisier than the unnormalized ones and we therefore do not report them.

**Table 1.** Subjective ratings and demographic variables and correlations between partners across dyads.

|          | relatedness | enjoyment | choice   | pressure  | effort    | ACIPS       | AQ         | age        |
| -------- | ----------- | --------- | -------- | --------- | --------- | ----------- | ---------- | ---------- |
| mean ± s.d. | 5.2 ± 0.9 | 5.8 ± 0.9 | 4.3 ± 1.2 | 2.2 ± 0.7 | 4.7 ± 1.1 | 85.1 ± 9.3 | 15.5 ± 5.1 | 25.9 ± 5.6 |
| min/max  | 2.5/6.9     | 3.0/7.0   | 1.8/6.6  | 1.0/3.6   | 2.3/6.8   | 52.0/101.0  | 7.0/28.0   | 20.0/50.0  |
| r        | 0.61        | 0.34      | 0.19     | −0.28     | 0.08      | 0.14        | −0.08      | 0.57       |
| p        | 0.0077      | 0.1661    | 0.4540   | 0.2574    | 0.7476    | 0.5739      | 0.7459     | 0.0135     |

trial of each experimental condition. Distances were calculated in two ways, either (i) between the location estimates of both participants as they were tracked by Kinect 2, or (ii) locations based on different Kinects for each participant after co-registration to a common reference frame. We report only the results of the two-Kinect approach, because this yielded more data and the results were consistent between the two approaches.

### 2.6.4. Interpersonal synchrony

For quantifying the interpersonal synchrony allowing for variable, non-zero time lags between participants, we employed a windowed cross-correlation [37] analysis approach. The cross-correlations were calculated in a 30 s (300-sample) sliding window moving in steps of 10 samples (1 s). Maximum lag allowed in the cross-correlation was 15 s in either direction. These data were averaged to produce a single mean windowed cross-correlation function for each condition for each dyad (note that this is different from the full cross-correlation over the entire data). The mean cross-correlation functions were then compared between conditions at each temporal lag. To summarize the interpersonal synchrony for correlating it with the subject scores, we used two approaches: (i) mean linear zero-lag Pearson correlation (mid-point of the cross-correlation), and (ii) a peak-picking algorithm to find the first positive peak in the mean cross-correlation functions closest to zero-lag inspired by the procedure suggested by Boker *et al*. [37]. These were then tested for correlation with behavioural and demographic variables. Null distribution of cross-correlation values was calculated identically, except the dyads and trials were shuffled randomly so that neither the participant nor the trial number matched in the randomized dyads.

## 3. Results

### 3.1. Behavioural ratings and personality scores

First, we evaluated the subjective ratings of the interactions (subject impression questionnaire of the intrinsic motivation inventory; [32]) and personality scores (AQ and ACIPS; the means and standard deviations of the individual scores and linear correlations of the interaction partners' ratings across dyads are summarized in table 1). The ratings revealed that most participants enjoyed the interactions (*enjoyment)* and felt related (*relatedness*) to the interaction partner. They did not feel pressured (*pressure*), but most still reported putting *effort* into the interaction, presumably due to the challenging nature of the gameplay task. Subjective feeling of *choice* in the interactions varied more widely, although most participants rated it relatively high.

The ratings of *relatedness* between interaction partners correlated significantly between dyads ($r = 0.61$, $p = 0.0077$), suggesting that feeling of relatedness was mutual in the dyads. Other evaluations did not show such agreement because of the relatively low variability of ratings between dyads. Age was also correlated between dyads due to one older dyad ($r = 0.57$, $p = 0.014$).

Subjective ratings of *relatedness* correlated significantly (two-tailed $p < 0.05$, Bonferroni corrected) with the subjective *enjoyment* and the ACIPS scores indexing the participants' traits for social/interpersonal pleasure (table 2). Correlations of *relatedness* with perceived *choice*, *enjoyment* with ACIPS scores, and negative correlation of ACIPS and AQ were significant only at an uncorrected threshold. However, neither AQ nor difference of AQ scores of the dyads predicted the behavioural ratings suggesting that the ACIPS may be a better predictor of social traits in healthy individuals.

**Table 2.** Correlations between the dyadic average ratings, personality questionnaires and age. $^{**}p < 0.05$, Bonferroni corrected; $^{*}p < 0.05$, uncorrected.

|  | relatedness | enjoyment | choice | pressure | effort | ACIPS | AQ |
|---|---|---|---|---|---|---|---|
| enjoyment | 0.901** |  |  |  |  |  |  |
| choice | 0.488* | 0.443 |  |  |  |  |  |
| pressure | −0.124 | −0.109 | −0.289 |  |  |  |  |
| effort | −0.277 | −0.247 | −0.381 | 0.003 |  |  |  |
| ACIPS | 0.697** | 0.620* | 0.121 | 0.133 | −0.143 |  |  |
| AQ | −0.241 | −0.172 | 0.207 | 0.142 | 0.35 | −0.487* |  |
| age | 0.004 | 0.176 | −0.24 | 0.013 | −0.194 | −0.176 | 0.006 |

## 3.2. Validating the optical motion tracking system

Initially, to verify the reliability of the optical motion tracking system, we compared how well the tracking results of the sensors corresponded to each other in all 18 dyads. We observed that the location estimates differed by approximately 3 cm, which stayed relatively constant over time (for the majority of the time, the error was less than 5 cm, and it stayed less than 10 cm throughout the experiment; see electronic supplementary material, results and figure S1 for validation results). We also observed that the upper body, head and dominant arm were similarly tracked by both trackers. By contrast, the lower body and non-dominant hand of the participants were not visible to the sensors due to visual occlusion by the table and were not tracked. Thus, we focus our analyses on the upper body, head and dominant hands in the remainder of the paper. The orientation of the head also corresponded well with the data recorded by the compass of the wearable IMU sensors in most participants. The correspondence was particularly high for participant 2 of each dyad, while results for participant 1 were more variable, presumably because they were located further away from the sensor tracking their movements. Estimates of acceleration of the dominant hand were also consistent between optical and wearable sensors, but the correlations were lower than for orientation data. This was presumably due to different sampling rate and increased sensitivity of the wearable sensor to small scale accelerations (see [26] for comparison between Kinect and Polhemus motion trackers that shows a similar effect for small-scale, but not large-scale movements).

## 3.3. Facial orienting behaviour

To study facial orienting and gaze behaviour, we started with two human observers rating a subset of randomly selected video frames during each of the tasks to identify joint gaze behaviours for each dyad: *eye contact*, *joint attention*, *other-oriented gaze* or looking *away*. The ratings were highly reliable, with the two raters agreeing on 96.8% of the individual ratings (looking at the partner, target or neither). For the time points where the raters disagreed, a third rater was recruited to find a consensus.

The relational yaw angles estimated by the Kinect sensors during each of the rated time points are visualized in figure 2*a–d* as coloured dots (red, *eye contact*, blue, *joint attention*, yellow, *other-oriented gaze*, grey, *away*). As can be seen, *eye contact* is mostly associated with low yaw angles for both participants (bottom-left of the plot) while joint attention is more spread out in the middle of the plot. Looking away corresponds to high angles whereas other-oriented gaze mostly occupies the centre of the plot (intermediate angles). We used these labels to train a classifier on the two-dimensional yaw space. The probability of each class of gaze behaviour estimated by the classifier is visualized as a coloured background behind dots representing the training samples in figure 2*a*. This classifier successfully recognized *eye contact* in the region indicated by a red outline (accuracy approx. 73%). The data of individual conditions (figure 2*b–d*) show that conversation trials consist mostly of *eye contact* and *other-oriented gaze*, but also include moments of the gaze category *away*, where at least one of the participants looked away from the table and the communication partner. By contrast, the gameplay trials consist almost entirely of joint attention as the participants look at the stick bundle together. Importantly, as an indication of the improved sensitivity of the dyadic joint orienting information, although the point clouds representing different gaze behaviours appear relatively

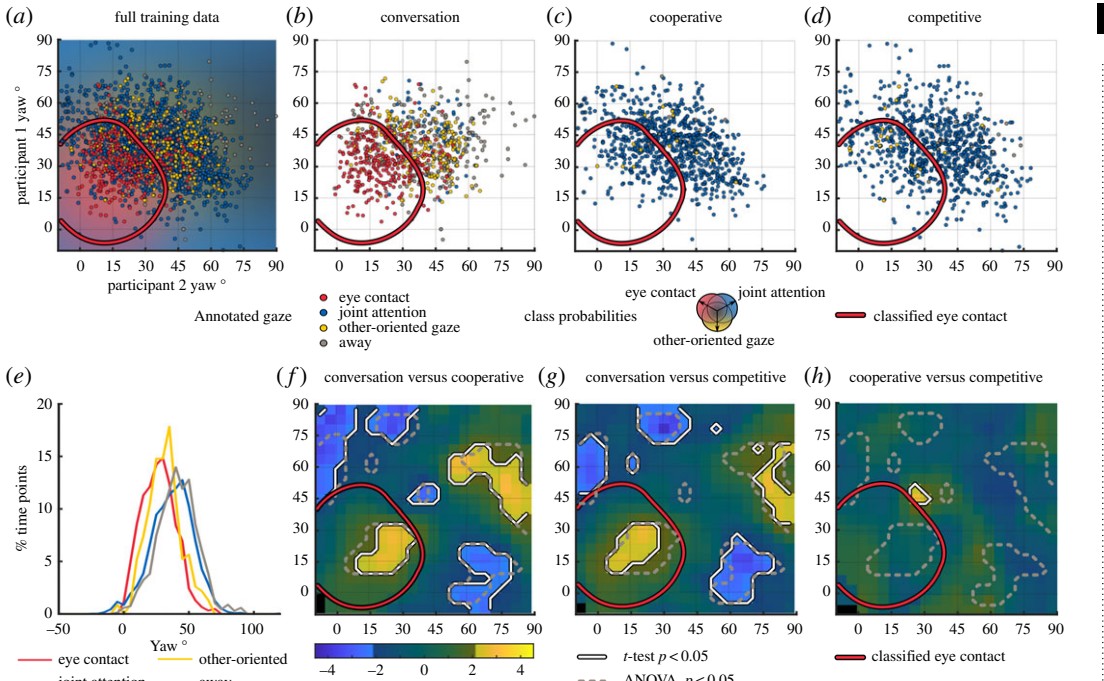

**Figure 2.** Gaze behaviours in the relational face orientation space. (*a*) Full training data showing the relational face orientations (Yaw$_1$ versus Yaw$_2$) of both participants colour coded based on the manually annotated gaze behaviours (red, *eye contact*; blue, *joint attention*; yellow, *other-oriented gaze*; grey, *away*). These data are overlaid on a colourmap showing the mean class probabilities of classifiers trained on balanced subsets of the manually annotated yaw data. Based on only the yaw information, eye contact is well-classified when both yaw angles are low (red contour; for comparison of classification accuracies, see electronic supplementary material, figure S2). (*b–d*) Scatter plots show the relational face orientation of the participants during manually labelled frames for each condition. Conversation moments contain a mix of all gaze categories, while gameplay conditions consist almost exclusively of *joint attention*. (*e*) Distributions of individual gaze angles in each gaze category in the training data. Individual gaze angle distributions overlap extensively. (*f–h*) Contrasts of normalized durations of relative face orientations between conditions over the entire dataset. Significant main effect of condition in the ANOVA is indicated by grey dashed outlines. Significant differences in pairwise *t*-tests between conditions are indicated by brighter colours and black-and-white outlines. Non-significant regions are shown with a muted colourmap. Region classified as eye contact is indicated by the red contour.

distinct in the dyadic orienting plots, there is considerable overlap between the distributions of gaze angles belonging to the different classes when participants' face angles are evaluated individually (figure 2*e*). However, even using the joint orienting data based on only the yaw angles of the face,[3] the moments of *joint attention* spread out considerably to regions that would intuitively be associated with *other-oriented* gaze (figure 1*c*). This seems to be related to large-scale movements during gameplay as participants move around the game bundle to get a better view of the individual sticks.

Next, we verified that the clear visually observed differences in orienting behaviour were also present in the full data when tested statistically (figure 2*f–h*). Using repeated measures ANOVAs on the smoothed (Gaussian kernel, $\sigma = 5°$) gaze heatmaps, we observed significant main effect of condition in several parts of the joint orienting space, shown by grey dashed lines in figure 2. Evaluating the underlying effects more thoroughly in paired *t*-tests between conditions, we observed increased joint orienting towards the partner during conversation versus gameplay trials ($p < 0.05$, uncorrected), corroborating our visual observation of increased eye contact during conversation trials. This region was also labelled as eye contact by the classifier (red outline; other classes were not well-recognized based on yaw angles alone). Increases during conversation versus gameplay were also seen in joint orienting away from the partner (high angles for both participants, usually associated with the category *away* during conversation trials). The opposite was true near the top-left and bottom-right

---

[3]We additionally employed the relational angular distance of gaze as well as yaw and combined (mean) pitch data of the two subjects for classification. However, the accuracies remained at approximately the same level as with yaw data alone and only improved when both individual yaw and pitch angles were included. Therefore, we visualize the relational orientations with only yaw data, which is the most intuitive to understand.

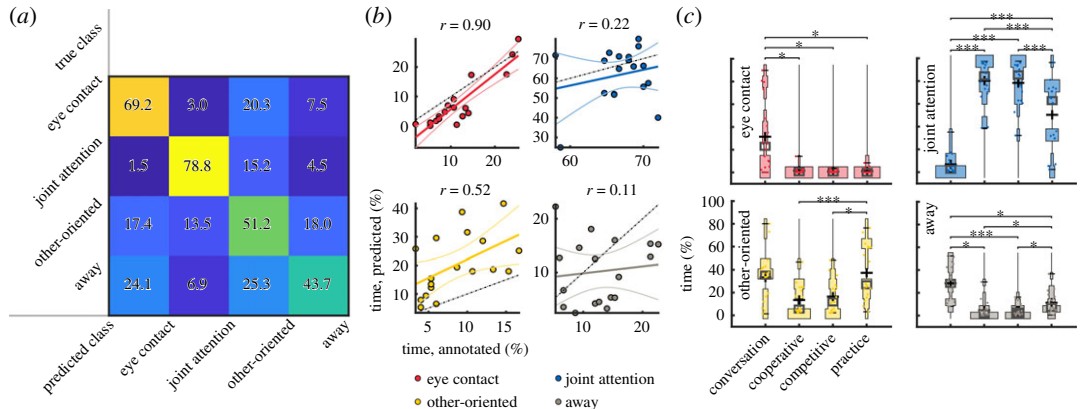

**Figure 3.** Classified gaze behaviours. (a) Mean confusion matrix showing the accuracies for each class (diagonal). The rows of the matrix show which classes the true classes are confused with. (b) Correspondence between annotated and predicted proportion gaze behaviours. Data for participants are shown as coloured dots and linear fit with confidence intervals between the annotated (partial data) and predicted (full data) behaviours is shown as coloured lines. Colours as in figure 2. One-to-one correspondence between predicted and annotated data is shown as black dot-dashed line. (c) Durations (%) of classified gaze behaviours during conversation and gameplay conditions. Significant differences between conditions are indicated by asterisks (*$p < 0.05$, ***$p < 0.001$, Bonferroni corrected).

corners corresponding to low face angles for one participant and high angles for their partner. The only difference that was observed between the cooperative and competitive rounds was a slight increase in joint intermediate face orientations at the edge of *eye contact* region (middle of the plot) in the cooperative versus competitive condition.

## 3.4. Predicting gaze behaviour from face orientation and location

To automatically detect gaze behaviours during the interactions, we employed the head yaw, pitch and location and the gaze (unit) vector as features in a four-class naive Bayes classifier. This set of features showed the best performance compared with reduced sets of features (electronic supplementary material, figure S2) and thus we focus only on this best-performing model in the following. Figure 3a shows the confusion matrix between the identified gaze behaviours. While all behaviours are recognized above the chance level (minimum accuracy approx. 44% for category *away*, chance level 25%), joint attention and eye contact are particularly well recognized. Moreover, the prevalence of the gaze behaviours over the entire data (in percentage of time; figure 3b) were highly correlated with the manually annotated amounts for *eye contact* and *other-oriented* gaze in the training set. The predicted amount of *joint attention* and *away* categories were more variable between participants, which seems to be exacerbated by one outlier (top left corner in the *away* category). The linear fits of *eye contact*, *joint attention* and *other-oriented* gaze follow closely the annotated amounts of these behaviours with slight under- (*eye contact*, *joint attention*) and over-estimation (*other-oriented*). These results indicate that some gaze behaviours can be classified from head data alone, suggesting that we can employ the predicted labels to compare gaze behaviours between conditions and across individuals. This is especially true for *eye contact*, which was of particular interest in the current study.

Using the prevalence of the predicted gaze behaviours (figure 3c), we observed highly significant interaction between task and gaze behaviour type ($F = 40.1$, $p = 6.4 \times 10^{-36}$). In pairwise comparison of specific gaze behaviours between conditions, we observed significantly more *joint attention* in all gameplay conditions than during the conversations. The opposite was true for *eye contact* and *away* categories. No differences were seen between the cooperative and competitive gameplay types. However, practice round of the gameplay, when participants jointly learned to play the game, consisted of more *other-oriented gaze* and *away* categories compared with the *cooperative* gameplay conditions. By contrast, less *joint attention* was observed in the practice compared with both *competitive* and *cooperative* gameplay conditions. We saw no significant changes in the orienting or gaze behaviour between the first and second conversation condition.

Here, we focused on predicting the amount of each gaze behaviour in the full data based on a small manually labelled subsample to see whether the face orientation could be used as a proxy for gaze of the participants. While we observed encouraging results, especially for eye contact, in some applications, it

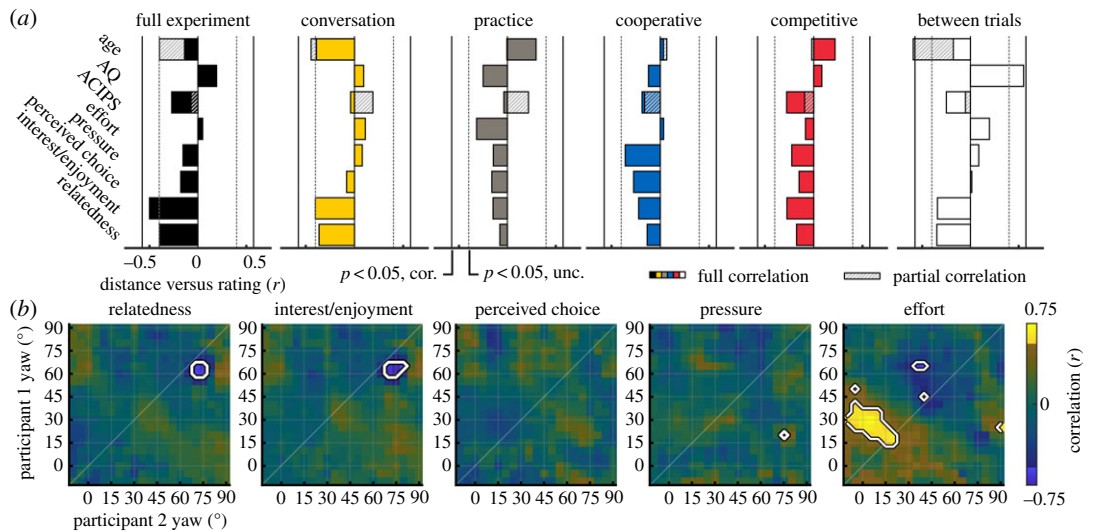

**Figure 4.** Correlations of subject scores and proxemic behaviour. (*a*) Correlations of interpersonal distance with subjective ratings and demographic variables across participants. Partial correlations controlling for outliers are indicated by hatched bars. The vertical lines indicate the level of significant correlation (dashed: $p < 0.05$, uncorrected; solid: $p < 0.05$, Bonferroni corrected by the number of scores, one-tailed). (*b*) Correlations of duration of joint orienting and subjective ratings of the interactions. Significant differences ($p < 0.05$, Bonferroni corrected by the number of scores, one-tailed) are indicated by brighter colours and black-and-white outlines. Non-significant regions are shown with a muted colour map.

would be preferable to be able to predict the gaze in unseen participants. We tested this with a leave-one-dyad-out cross-validation and observed a mean accuracy of approximately 57% (null max. 26.9%) across gaze categories and dyads, compared with the approximately 61% mean accuracy (null max. 29.7%) in the randomly split data. While the performance was still well above chance level, variability across dyads was relatively high. This was particularly true for eye contact, which was not observed at all for some participants (figure 3*c*) leading to 0% accuracies for the category and thus reducing the mean across dyads.

## 3.5. Proxemic behaviour as a predictor of behavioural ratings and personality scores

To relate the participants' orienting and movement behaviour to their subjective evaluations of the interactions, we correlated the interaction ratings and individual characteristics (age, AQ, ACIPS) with the proxemic measures between participants. The positive ratings of *interest/enjoyment* correlated negatively ($r = 0.498$, one-tailed $p = 0.0177$) with interpersonal distance over the entire experiment (figure 4*a*). In other words, dyads reporting higher mean enjoyment sat (or leaned) closer to each other. Additionally, correlations with *relatedness* were near significant ($r = -0.3947$, $p = 0.0525$) as might be expected due to high similarity of *enjoyment* and *relatedness* ratings (table 2). At the trial level, these effects were not significant, but remained near the threshold during the *conversation* moments ($r_{interest} = 0.40$ $p = 0.05$, $r_{enjoyment} = 0.36$, $p = 0.07$) and pauses *between trials* ($r_{interest} = 0.34$ $p = 0.08$, $r_{enjoyment} = 0.35$, $p = 0.08$), when experimental control was minimal. During gameplay these correlations were low. Additionally, distance between participants correlated positively ($p \sim 0.009$) with the mean autism quotient of the dyad, but only during the pauses between trials. Due to outliers in the age and ACIPS data (defined as dyads three standard deviations away from the mean), we also calculated partial correlations with these measures controlling for the outliers. In this analysis, we saw an additional negative correlation of age with distance (decreased distance with increasing age in young adults) during unconstrained moments of *conversation* ($p \sim 0.033$) and *between trials* ($p \sim 0.005$), which was also nearly significant ($p \sim 0.055$) during conversation in the full correlations.

The amount of effort participants reported investing in the interaction was associated with orienting more towards the interaction partner (lower relative gaze angle) at the individual level ($r = -0.44$, $p < 0.05$ Bonferroni corrected for the number of ratings). However, analysing the data at the level of the dyads revealed a stronger correlation ($r \sim 0.7$) of *effort* and orienting behaviour that was specific to the time participants spent jointly orienting towards each other (putative eye contact; figure 4*b*) rather than a simple global shift in yaw angles. Largest effects were seen during the cooperative gameplay trials, but correlations were also significant during competitive trials.

To explore whether the observed correlation of subjective ratings and orienting behaviour was specific to the gaze behaviour categories, we correlated the amount of the gaze behaviours predicted by the classifiers during each trial with the subjective ratings and personality variables. Using the classifier based on only the yaw information over the entire experiment, we observed a significant correlation of *eye contact* and *effort* ($r \sim 0.66$, $p \sim 0.0027$) similar to figure 4*b*. However, when using the most successful classifier (figure 3), this correlation was no longer significant, suggesting that mere joint facial orienting in the direction of the interaction partner, rather than *eye contact per se*, was predictive of the rated *effort*.

Body and face directions of participants were largely similar over the experiment and the correlation of *effort* and joint orienting behaviour shown for the face was also observed for body orientation (electronic supplementary material, figure S3), although shifted towards larger angles for participant 1. The body was typically oriented to a lesser degree towards the interaction partner than the face, but generally, the face and body directions were correlated over time for each participant (see electronic supplementary material, figure S4 for visualization of face versus body angles for all dyads).

## 3.6. Interpersonal synchrony of head and hand movements

Finally, to evaluate the interpersonal synchrony between the participants, we calculated windowed cross-correlations [37] of the movement speed of the heads and the dominant hands for each dyad. To summarize the most consistent synchrony over each condition, we calculated the mean cross-correlation over time windows of all trials of each type. The mean cross-correlation functions over dyads (±95% confidence interval) and their differences between conditions are depicted in figure 5 (upper and lower triangles for head and hand synchrony, respectively). Comparing the cross-correlation functions of head speed between conditions, we observed significant ($p < 0.05$) main effect of condition in 52.5% of all time delays (dark grey bars in figure 5). In the pairwise comparisons between conditions, we see that the peak of cross-correlation around zero-lag is wider in the conversation condition than gameplay conditions (tails of the peak are significantly higher during conversation than *cooperative* or *competitive* gameplay, but not compared with *practice*). This presumably reflects more flexible timing of turn taking and mimicry during conversation. Conversely, gameplay conditions exhibit additional peaks in cross-correlation at lags of several seconds, presumably corresponding to the pace of the gameplay. Peak locations indicate that the pace is faster during practice (peak distance approx. 5 s) than other gameplay rounds (approx. 10 s), where participants have to choose sticks for the other player and possibly negotiate the choice with them before choosing. These side-peaks do not exceed the 95th percentile in the *cooperative* condition, which is also true for majority of peaks in the *practice* condition, but the differences in the side-peaks remain non-significant between the gameplay conditions. However, the zero-lag correlations in the practice condition are significantly higher than *scored* gameplay conditions (*cooperative* and *competitive*).

In contrast to the results for head synchrony, the cross-correlations for hand speeds are strikingly similar in all conditions with a tight peak around zero-lag. ANOVAs showed significant main effects for condition in 14.95% of time points. The only significant differences in the pairwise comparisons are caused by negative cross-correlations in *conversation* and *practice* conditions coinciding with a tightly clustered zero-correlation in the cooperative condition at a lag of approximately 10 s.

Finally, to analyse the relation of interpersonal synchrony with behavioural ratings and characteristics of the participants, we extracted the mean cross-correlations at zero-lag (normal linear correlation) and at the first positive peak in the cross-correlation function near zero-lag for each dyad (crosses in figure 5; see Data analysis section). We then calculated the correlations between these synchrony values and the subject scores. Additionally, we calculated partial correlations controlling for outliers for age and ACIPS (see above). Overall, we see a repeating pattern of increased interpersonal synchrony with age, although it is not significant in all conditions and is partly driven by one significantly older dyad. However, even after controlling for the outlier, the effect remains significant in the *cooperative* condition and becomes significant in the *conversation* condition. The only other correlations that are significant for both peak and zero-lag synchrony are negative correlations of hand synchrony and choice in conversation and cooperative conditions, and positive, threshold-level effects of head synchrony–pressure and hand synchrony–effort correlations during practice gameplay, but not other conditions.

## 4. Discussion

Our results demonstrate that optical motion tracking can be used for non-invasive tracking of natural, and relatively unconstrained social interactions of extended durations to extract objective, compact and

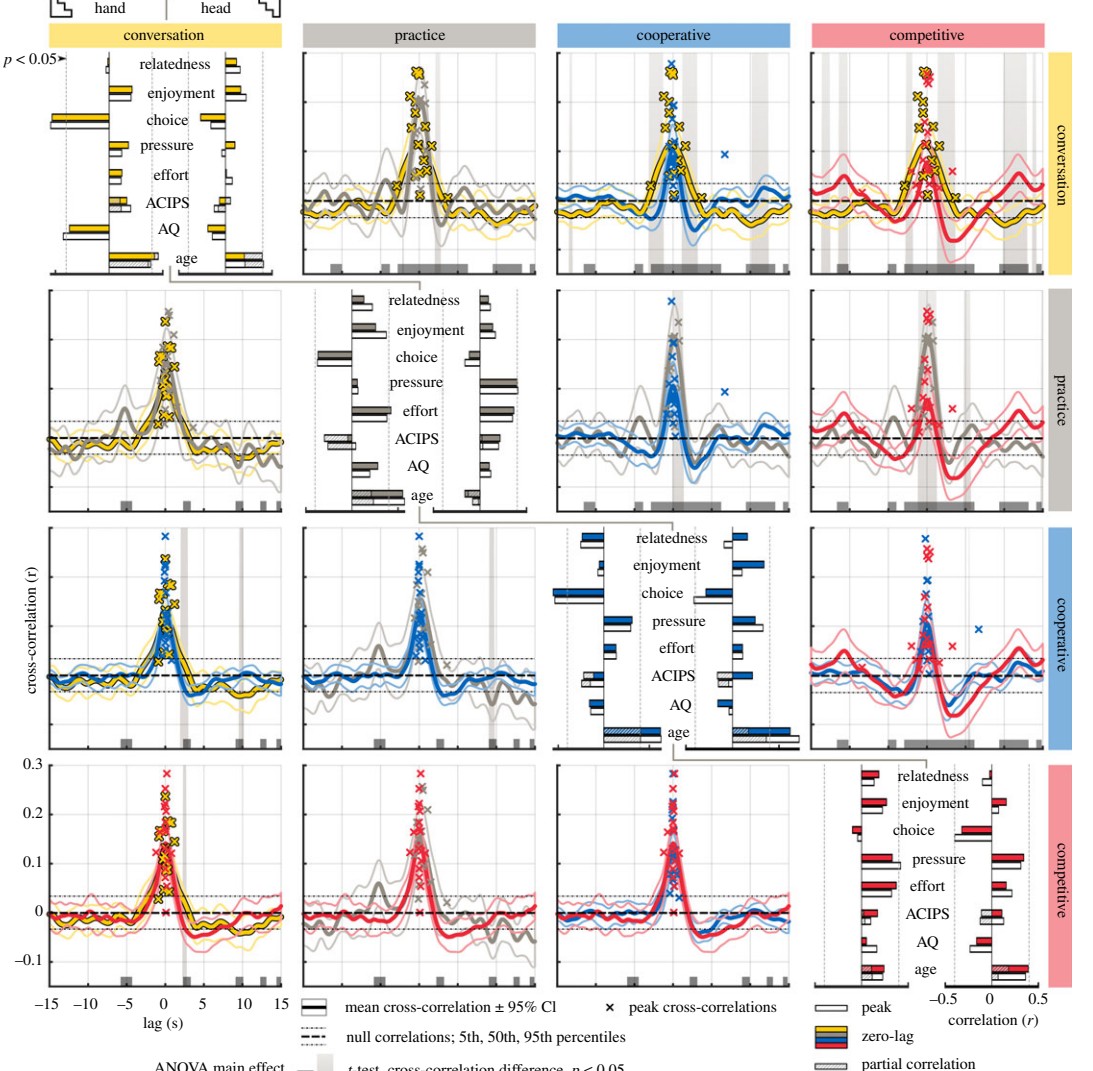

**Figure 5.** Behavioural synchrony differences for heads (upper triangle) and hands (lower triangle) and correlations of behavioural synchrony with subject scores (diagonal: left, hand; right, head). Upper/lower triangle: each plot depicts the mean windowed cross-correlation functions within dyads over all time windows of all corresponding trials (mean ± 95% confidence interval over dyads) for two conditions. Significant main effects for condition are shown as low dark grey bars (ANOVA) and differences between pairs of conditions are indicated by grey background (t-test). Percentiles of null data are indicated by black dashed lines. The location and height of the first positive peak of the cross-correlation function nearest to zero-lag is indicated by the coloured crosses for each dyad. Scale for all plots is identical and is indicated on the bottom left. Diagonal: correlations of behavioural synchrony with subject scores. Coloured bars indicate the correlations at zero-lag and white bars indicate the correlations with the peak correlation. The hatched bars indicate partial correlations controlling for outliers. Threshold for significant correlation ($p < 0.05$, uncorrected) is indicated by the dashed grey lines. Scale for all plots is indicated on the bottom right.

interpretable measures of interpersonal behaviour. These measures build on traditional measures of interpersonal synchrony to provide a more extensive characterization of interpersonal behavioural patterns that are predictive of the subjective experiences of the interactants.

Specifically, the participants' joint orienting behaviour towards their partners while they played a social game was indicative of the self-reported effort they put into the interactions, particularly during cooperative trials. A similar effect was also present for the direction of the upper body, although less prominently. Moreover, the interpersonal distance of individuals during the most unconstrained moments of the interactions correlated negatively with the subjective *pleasure* and *relatedness* and positively with the autistic traits (AQ) of the participants, suggesting these measures could be of use in characterizing behavioural differences in psychiatric conditions, which are ubiquitously characterized by social impairments [38]. Importantly, using a machine learning approach, we could

distinguish between gaze and orienting behaviour without the use of explicit eye tracking; orienting (horizontally) towards the partner was correlated with reported effort, particularly during cooperative gameplay, while any of the gaze behaviours were not, suggesting that different information can be extracted from facial and bodily orienting compared with gaze. Contrary to our predictions, increased familiarity towards the end of the experiment did not significantly affect joint orienting or joint eye gaze measures, possibly due to the duration or design of the experiment, which may not have been optimal for investigating these effects. Surprisingly, even though we observed interpersonal synchrony during all of the tasks, as well as clear peaks of delayed synchrony corresponding to the pace of the repetitive gameplay task, interpersonal synchrony seemed less predictive of the subjective evaluations of the interaction than proxemic measures in the current context. These findings extend research employing more constrained interactions and simpler motion quantification methods by more accurate tracking of head orientation and fine-grained location tracking of specific body parts in a three-dimensional space. Our results further suggest that dyadic measures of social orienting and distancing behaviour may be more predictive of subjective evaluations of social interactions than the more widely employed synchrony measures in some situations.

The reliability of these results supports the use of optical body tracking for non-invasive quantification of patterns of social gaze, orienting and interpersonal synchrony. This could enable large amounts of data collected during unconstrained interactions to be summarized, potentially providing an observer-independent assessment of social abnormalities in individuals with psychiatric disorders. In the future, interaction-based motion tracking might, therefore, help to quantitatively address well-known clinical intuitions such as the 'praecox feeling' in schizophrenia [39] and reduced sense of social presence in autism [40]. While some evidence exists for disrupted gesturing during interactions in psychiatric disorders, such as schizophrenia [41], further work is needed to evaluate what improvements may still be needed to enable the objective, interaction-based phenotyping of patients.

## 4.1. Experimental and subjective correlates of proxemic behaviour

Our results show that proxemic behaviour including interpersonal distancing and facial and bodily orienting during natural interactions differs between types of interactions and reflects subsequent subjective evaluations of the interaction. This is in line with our general hypotheses that interpersonal behaviour is predictive of the quality of interaction and that the type of interaction affects the level to which interaction quality is reflected in the interpersonal measures. This was particularly true when comparing free conversation to the gameplay conditions, where not only behaviour was clearly different but also the level of correlation of behaviour and subjective evaluations were differentially sensitive to orienting and distancing.

Intriguingly, our results suggest that some prosocial proxemic behaviours, such as sitting closer to people with whom you enjoy interacting, might be most readily observed when experimental control is minimized. Correlations with enjoyment and relatedness were observed over the full experimental session, but they were driven by conversation trials and pauses between trials with no experimental task rather than moments of gameplay. Similarly, in the exploratory analysis focusing on the unconstrained pauses between trials when participants sat neutrally without an explicit need to interact, autistic traits measured by the AQ were correlated with increased distance suggesting that proxemic patterns associated with autistic traits might be most easily observed when people interact freely with no explicit task. This is also in line with clinical insights and procedures for testing autistic behaviour (such as the Autism Diagnostic Observation Schedule; ADOS [42]). However, this observation should be tested more thoroughly in future studies. Specifically, the gameplay tasks employed here imposed relatively high task demands to the movement of the participant because the instructions dictated a relatively strict turn-taking behaviour. Moreover, successful gameplay also required considerable large-scale movements of the upper body as the participants moved around the bundle trying to select the optimal stick for the particular condition. These behavioural requirements imposed by the task may have masked some of the subtle differences in interpersonal proxemics behaviour that were observed during the discussions and pauses when the posture was more static. However, while the data suggest that participants returned to their neutral sitting position between the pauses, the actions of the interviewer may have had an effect on their body position making the conclusions on this data tentative.

Unlike correlations between distance and positive evaluations of the interaction, which were mainly observed during conversation and task-free moments, subjective reports of putting effort into the interaction were reflected in the joint orienting behaviour primarily during cooperative gameplay. Specifically, participants' face angle in the direction of their partner (analysed individually) as well as

the joint orienting behaviour towards the partner correlated with the subjective effort mainly in the cooperative condition. Cooperation has previously been associated with coordinated gazing and orienting behaviour. For example, people tend to show more cooperative behaviours when they are watched [43,44], young children interpret communicative eye contact as a commitment for cooperation during gameplay [45], and joint gazing facilitates joint action such as timing during musical ensemble performances [46]. In the current study, orienting towards the partner was presumably associated with a mix of gaze behaviours (*other-oriented gaze* and *eye contact* and possibly *joint attention*) as we saw no correlations with any single gaze behaviour and subjective ratings.

Additionally, the orienting results for the upper body and the face were qualitatively similar in the current study, suggesting that the participants oriented not only their face, but also their body towards the partner. However, the angles of the body were higher (body was not oriented as directly towards the partner as the face) and effects on the body orientation were weaker. This is intuitively logical as it is easier to turn the head than the whole upper body and it is also unnecessary to turn the whole body to look at the partner or the target on the table. Indeed, the non-zero angles of the face during moments of eye contact (red dots in figure 2*a,b*) also suggest that people do not usually orient their face directly at the partner but rather also turn their eyes during eye contact, at least when sitting diagonally at a table. Therefore, even if the face direction and location predicted all gaze categories well above chance, only eye contact was detected at a level that strongly predicted the amount of the behaviour at an individual level, as suggested by the correlations of real and predicted gaze behaviour durations across dyads. Yet, even the accuracy of 70–80% attained for eye contact and joint attention means that the confidence for any single time point is not extremely high despite the classifier producing good congruence over the entire experiment. Here, our goal was to test whether we can predict gaze from the Kinect data to see whether labelling only a small subset of data would allow us to infer the gaze behaviour during the rest of the interaction. However, if the goal is to use the detection scheme as a predictor of gaze in unseen participants, as might be the case in clinical studies, the generalizability of the model across subjects should be improved from the approximately 57% mean accuracy achieved here. These imperfect prediction results are attributable not only to the imperfect correspondence of eye and face directions and interindividual differences, but also to the relatively noisy head orientation data. Thus, further work should seek to improve both data quality and prediction accuracy, which may be achieved by future improvements to the tracking hardware itself as well as optimizing the positioning of the sensors in relation to the individuals being tracked and further development of motion tracking and classification algorithms. One additional aspect that may have explained the poorer performance in some classes of dyadic behaviour is the ambiguity of the gaze categories themselves. In particular, some time points in the *away* category contained moments when only one participant was looking away while their partner was looking at them, which could have equally well been categorized as *other-oriented gaze* had we defined the categories differently. This definition was motivated by the behavioural significance of either person looking away from the other as well as the low number of samples where both participants were looking away at the same time (approx. 2% of training data; see electronic supplementary material, table S1). However, this may have made these two classes more heterogeneous and ambiguous than the others, thereby making them more difficult to classify.

## 4.2. Interpersonal synchrony and similarity of movement profiles

Interpersonal synchrony is observed at several levels, from synchrony of overt actions to physiological signals such as breathing and heartbeat to electrical and haemodynamic brain activity (for a brief overview, see [14]). At the behavioural level, synchrony and unconscious imitation have been linked to positive outcomes, such as success of psychotherapy [47] and increased affiliation [48]. It is, therefore, a promising target of investigation to quantitatively assess both successful and disordered social interactions.

In the current study, we employed a windowed cross-correlation analysis to assess the interpersonal synchrony allowing for variable time lags introduced, for example, by the turn-taking behaviour during the gameplay trials. We observed significant synchrony between both the heads and hands of the participants compared with null data where the dyads and trials were randomly shuffled (figure 5, top and bottom triangles, respectively). Similarly to the results on interpersonal distance, the less controlled parts of the experiment, namely the conversation and practice tasks, showed the highest average synchrony, similar to the correlations between orienting and behavioural ratings discussed above. Specifically, the peak of the cross-correlations was wider during the conversation condition

than the scored gameplay rounds suggesting more flexibility of behavioural synchrony when task was less controlled. Additionally, the cross-correlation peak was significantly higher during *practice* than *scored* gameplay rounds, suggesting people may have synchronized with their partners' behaviour more closely when they were jointly learning the task rather than taking turns performing it. Moreover, as a validation of the method, we also observed the expected side-peaks of synchrony during gameplay conditions, which presumably correspond to the repetitive nature of the task, and which were not visible in the *conversation* condition. The distance between these peaks was higher during *scored* gameplay versus *practice* rounds, confirming that it took (on average) more time to choose the appropriate stick for the other person than to pick it out oneself during unconstrained practice. In the *cooperative* condition, the side-peaks remained below the 95th percentile of the null distribution, although differences between gameplay conditions were not significant. These low peaks may be explained by additional verbal negotiation between participants on the best stick to choose during cooperation causing more variable turn durations. However, despite subtle changes, our expectation of clear differences between competitive and cooperative conditions was not fulfilled either in the synchrony or the proxemics measures. This suggests that the tasks were indeed well balanced in overt behaviour, which was a goal of the task design, but more sensitive measures should be designed to detect differences in the underlying motivation of the participants in the two gameplay conditions.

Despite observing clear behavioural synchrony in all task conditions, our hypothesis that quality of interaction would be reflected as increased interpersonal synchrony was largely not supported by the current data. Behavioural synchrony correlations of participants' ratings of the interactions were largely not significant (figure 5, diagonal), suggesting that behavioural synchrony may be less sensitive to these types of differences than proxemic measures. Recently, Hale & Hamilton [15,16] demonstrated that the relationship between mimicry and prosocial outcomes may not be robust in all conditions leading to variable findings, at least in a virtual reality setting, which may explain some of the negative findings here. Ratings of pressure and effort did correlate with increased synchrony during practice, although this did not repeat in other conditions, including the cooperative condition, which seemed most sensitive to effects of effort in the joint orienting data.

While correlations with behavioural ratings were mostly not significant, we saw a repeating pattern of age correlating with increased interpersonal synchrony, which was affected but not abolished by removing one significantly older outlier dyad. Intriguingly, after removing the outlier, we also saw a negative correlation of distance and age (figure 4*a*). While distance was also negatively correlated with positive evaluations of the interactions, correlations of age with positive interaction ratings as well as with ACIPS scores were non-significant and negative (table 2). The correlations of these measures with interpersonal synchrony also remained negative during all conditions. Thus, positive evaluations and age may independently affect the comfortable distance between communication partners. While age is often considered a variable of no interest within an age group, our results suggest that interesting interactions may exist in (early) adulthood between age, subjective evaluations of interaction success, interpersonal synchrony and proxemic behaviour. However, due to the limited sample size in the current study, these results should be explored further in an independent sample and experimental paradigm specifically designed to test the observations shown here. These and other limitations are discussed in greater detail in the electronic supplementary material, discussion.

## 4.3. Limitations and future directions

Bodily motion can also reveal aspects of the emotional state of an individual that are readily detectable by human observers, even from extremely simplified point-light displays of human motion without seeing the shape of the body (e.g. [49,50]). Machine learning methods have also made it possible to use such information to computationally detect the emotional state [51] and laughter [52] of individuals based on their motion tracked by the Kinect, which could be beneficial for deeper understanding of the kind of behavioural differences reported here. Thus, expanding the analyses here with emotion detection may be an interesting avenue for future research.

While the current motion tracking system is useful for tracking natural face-to-face social interactions, such as those between a client and therapist during a diagnostic interview or therapy session, it is limited to relatively small and static spatial locations because of the range and field of view of the tracking system. To address more unconstrained daily activities, wearable technologies can extend measurements to real-life conditions. For example, radio frequency identification tags have been used to measure simple aspects of interactions, such as number and length or interactions in freely

interacting individuals at a workplace break room [53] or in a conference setting [54] to evaluate social interaction patterns between individuals. With the advent of smartphones, digital phenotyping [55] has allowed measuring things like movement and digitally mediated social interactions in everyday life with the participants' own personal devices. However, most wearable devices do not allow a detailed characterization of subtle forms of non-verbal communication such as bodily and facial gesturing, which may offer important information about the nature of interactions. Therefore, these approaches can reveal complementary information and the most fruitful approaches of future research may be a combination of detailed body tracking during relatively static interactions combined with a day-to-day characterization of daily life at a larger scale using wearable devices or smartphones.

Finally, we saw no clear effects of familiarity between the interaction partners when comparing the conversation tasks at the beginning and end of the experiment. This could mean either that participants did not show a clear increase in familiarity during the relatively short experiment or, alternatively, that the methods employed here are not sensitive to subtle effects that may have arisen. Future studies should induce a stronger sense of familiarity between participants, for example, through repeated social interactions across longer time periods as is the case in therapeutic setting, to directly test these hypotheses.

## 5. Conclusion

Our results show that dyadic analyses of unobtrusive motion tracking data obtained from freely interacting dyads provide robust, objective measures of both individual and interpersonal movement patterns. These measures are predictive of subjective evaluations of the social interaction and interaction types and are validated by a wearable motion tracking system. In particular, our machine learning results suggest that facial and bodily orienting contain information that is predictive of subjective evaluations of the interaction, in this case even more so than specific gaze behaviours. Future work may extend these analyses into patient groups with difficulties or disorders of social interaction (such as autism, schizophrenia, personality and anxiety disorders and depression) to find sensitive predictors that may be of use in a clinical context.

## Significance statement

Interpersonal coordination of behaviour is essential for smooth social interactions. Measures of interpersonal behaviour, however, often rely on subjective evaluations, invasive measurement techniques or gross measures of motion. Here, we constructed an unobtrusive motion tracking system that objectively measures interpersonal orienting, gaze and distancing behaviour, which we validated using wearable sensors. Results demonstrate that patterns of proxemic behaviours, rather than more widely used measures of interpersonal synchrony, best predicted the subjective quality of the interactions. Our results also suggest that natural, unconstrained interactions with minimal experimental interference may be the best conditions for revealing behavioural differences that are related to interindividual differences in sociability and interaction success. The approaches demonstrated here may, thus, be useful in developing objective measures to quantify and predict differences in and disorders of social interaction as seen in psychiatric conditions.

Ethics. The experimental procedure followed during the study was accepted by the ethics committee of the Ludwig-Maximilians-University Munich (approval number 624-16). All participants gave written informed consent before participating in the study and the participants were compensated for their time.

Data accessibility. All the code and the anonymous motion tracking results to produce the final results in this manuscript are available from the Data Dryad Repository of this article (https://doi.org/10.5061/dryad.v9s4mw6sp) [56]. Due to privacy concerns and limitations in the ethics permit, the raw data, such as the video frames that could identify individual participants, cannot be made publicly available.

Author contributions. Designed the study: J.M.L., P.A.G.F, L.S.; gathered the data: P.A.G.F; oversaw data acquisition: J.M.L.; planned the data analysis: all authors; analysed the data: J.M.L.; wrote the manuscript: J.M.L., P.A.G.F; edited the manuscript: all authors; oversaw the project: L.S.

Competing interests. We declare we have no competing interests.

Funding. This work was supported by Max Planck Society via an Independent Max Planck Research Group (L.S.), Finnish Cultural Foundation (grant no. 150496 to J.M.L.), and the Experimental Psychology Society (grant to P.A.G.F. for a research visit at the MPIP).

Acknowledgements. We thank Marie-Luise Brandi for suggestions on the experimental design and Dimitris Bolis for fruitful discussions.

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
