## [Reviewer comments · Royal Society Open Science]

Review History

RSOS-191815.R0 (Original submission)

Review form: Reviewer 1

Is the manuscript scientifically sound in its present form?

No

Are the interpretations and conclusions justified by the results?

Yes

Is the language acceptable?

Yes

Do you have any ethical concerns with this paper?

No

Have you any concerns about statistical analyses in this paper?

Yes

Recommendation?

Major revision is needed (please make suggestions in comments)

Comments to the Author(s)

This paper reports on measures of interpersonal coordination and proxemics from a kinect motion sensor system which they compare to the measures provided by a wearable sensor. I am very sympathetic to this work, as the advent of kinect motion capture technology has the potential to revolutionise several fields, and knowing how accurate or otherwise the data obtained is is clearly necessary before the methodology becomes relied upon as if the gold-standard. However, there were a couple of issues that need to be addressed before this paper is published. Firstly, there were a few key references (addressing the same concerns as the papers authors) that are missing that should be referenced (e.g. Khoshelham and Elberink, 2012; Lavelle et al., 2012; Nichols et al., 2017). Secondly, the structure of the paper was somewhat confusing with the split of information between the methods and results not clearly delimited or signalled so that when first encountering information about, for example, the annotation scheme and classifiers this is incomplete and confusing and it was not clear to me exactly what you had done. Splitting the information between the method and results is fine, as long as the reader is made aware that the expected information is coming (details of the major confusions are provided in the 'specific comments' below). One other major concern I had is that the authors introduced a control condition (p.15 l.28; which is, in my view, a very good thing), but it is never clear exactly how this control condition is generated or whether/how this has been used in the analysis (I could not find any reference to it in the results, and in the discussion it claims it is what figure 5 shows, but this is not what the caption for figure 5 says).

In the results I have some questions about why you chose to use individuals as the unit of analysis for some sections, but both individuals and dyads for other sections. There are also issues about the reporting of significant effects between trials -- these periods are not mentioned in the methods as something you would be analysing which gives the reporting of them the character of a 'fishing expedition' (performing multiple statistical tests until you find a significant result, rather than performing specific tests to address specific hypotheses). These need to either be already part of the planned analysis from the start of the paper, or removed.

The discussion would be clearer if it directly addressed the three hypotheses brought up on page 8 (l.51).

Specific comments.

p.14 l.19 "For those timepoints where the two raters disagreed, a third rater was employed to break the tie." What was the frequency of this? What were raw kappas? (Aha, these are answered in the results: please make a note of this to guide the reader). Did the third annotator know the two tie-break options or choose their annotation based on all 4 options?

p.14 l.35 "Combinations of these data points (yaw, yaw+pitch, yaw+pitch+location, yaw+pitch+location+gaze vector) were used to train Naïve Bayesian classifiers to detect the four gaze categories using random balanced subsamples of the data." This is uninformative. Which did you use? What was the accuracy/recall or f-score? How do I know these were any good at performing this classification task? -- again, some of this comes up in the results (importantly not which classifier you ended up using, which is highly relevant, or the accuracies for other categories), but the reader needs to at least be aware it is coming. I don't actually know what you've done here.

Results: First section seems to be about individuals; have you compared how dyadic partners rated the interaction in relation to each other?

Table 2 (correlation matrix) would be clearer if repeated values were removed (i.e. those above the diagonal)

p.23 1.8 "At the trial level, these effects were only significant during the conversation moments and near significant during pauses between trials, when experimental control was minimal. Additionally, distance between participants correlated positively with the mean autism quotient of the dyad, but only during the pauses between trials." You've never even stated that you are going to look at pauses between trials, so it reads as if this findings are the results of statistical 'fishing'. Also, you need some numbers (how 'near significant'? $p=0.55$? $p=0.1$?)

p.24 1.5 "per say"  "per se"

References:

Khoshelham, Kourosh & Oude Elberink, Sander. (2012). Accuracy and Resolution of Kinect Depth Data for Indoor Mapping Applications. *Sensors* (Basel, Switzerland). 12. 1437-54. 10.3390/s120201437.

M Lavelle, PGT Healey, R McCabe (2012) Is nonverbal communication disrupted in interactions involving patients with schizophrenia? *Schizophrenia bulletin* 39 (5), 1150-1158

Nichols, Julia K., Sena, Mark P., Hu, Jennifer L., O'Reilly, Oliver M., Feeley, Brian T. and Lotz, Jeffrey C. (2017). A Kinect-based movement assessment system: marker position comparison to Vicon. *Computer Methods in Biomechanics and Biomedical Engineering* 20(12), 1289-1298, 10.1080/10255842.2017.1340464

Review form: Reviewer 2

Is the manuscript scientifically sound in its present form?

No

Are the interpretations and conclusions justified by the results?

No

Is the language acceptable?

Yes

Do you have any ethical concerns with this paper?

No

Have you any concerns about statistical analyses in this paper?

Yes

Recommendation?

Major revision is needed (please make suggestions in comments)

Comments to the Author(s)

The paper presents a low-cost movement sensor tracking approach to investigate the correlations between quality of social interaction experience and dyad's behaviour. Kinect 2 is used and evaluated against a more robust tracking system. New measures of dyad's behaviour are proposed based on the sensed data. These are complemented by eye-gazing manual annotations of random selected samples. Qualitative self-reports are used to characterise the social experience of the dyad. The results suggest that typical assumptions of interpersonal synchrony may not predict the quality of social interaction and instead patterns of proxemics behaviour may be more informative. The authors also claim that the results show the importance of reducing

experimental control to investigate these questions. Data (but not videos) and software are made available to the community.

The paper is interesting and timely as it explores the opportunity that sensing technology offer to really understand people behaviour in relation to experience in real-world more complex settings (or close to). The study is supported by the automatic and manual analysis of the behaviour of 10 dyads in different conditions. The paper is overall well written but the lack of detail makes the understanding of the results and soundness of the analysis difficult.

LITERATURE REVIEW

- The authors should provide a review of the studies on the use of Kinect, RFID or another low cost sensors to study social interaction qualities (e.g., Mancini et al., 2014 & Cattuto et al. 2010) and the features used to understand proxemics behaviour and joint attention using such sensors. They should also look at work in the area of affective computing aimed at detecting emotional responses in dyads using such movement sensors.
- A discussion on the selection of Kinect vs other type of sensors is also welcome as Kinect is quite limited in less static situations. This is particularly important given the accent made on the need for reduced experimental control.

SYSTEM VALIDATION

- Clarify how many participants were used for this validation.

DATA ANALYSIS

Detailed explanation of the data analysis and modelling should be provided. It is unclear what data are used in each of the analysis and if the process is sound.

- T-test: What data were used for the various t-test? Multiple extracts from the same dyads? What are the paired data? The data are clearly not independents and not paired related. Or just mean values over the full sessions? Why just t-test given that there are 3 conditions. Why not a more comprehensive test that take into account the different conditions?
- Classifiers: how are the classifiers trained and tested? Using training samples that are closely related to the testing samples (e.g., close by along the time dimension) lead to higher performances than using more time-unrelated data. Leave one subject out would be better to understand the generalization capabilities of the classifiers. Number of samples per class should be provided.

FINDINGS: the results from the classification are above chance levels but not very high for all classes. Using the classifiers to drive conclusions on what features are more relevant to understand behaviour quality may be risky. E.g., The fact that some gaze behaviour can be predicted by head data alone does not lead to the conclusion that the approach can be used for eye-gazed type prediction. Similarly, it is not possible to conclude on the correlation between classified eye-gaze type and quality of interaction as many eye-gaze types were misclassified. Such claims should be reduced or discussed in relation to the limitation of the approach.

DISCUSSION: it is unclear how the authors concludes that proxemics behaviour is more present when experimental control is minimized. Shouldn't the study consider a variety of contexts? This should be elaborated and grounded in the results. Differences in synchrony between the different conditions should be discussed in relation to the type of movement imposed by the specific tasks. The current analysis does not allow to take into account this factor.

Overall, the work is interesting but a more detailed description of the study and more critical analysis of the contribution made and the limitation of the features proposed should be provided.

Cattuto C, Van den Broeck W, Barrat A, Colizza V, Pinton JF, Vespignani A. Dynamics of person-to-person interactions from distributed RFID sensor networks. *PLoS One*. 2010;5(7):e11596. Published 2010 Jul 15. doi:10.1371/journal.pone.0011596

Mancini, M., Ach, L., Bantegnie, E., Baur, T., Berthouze, N., Datta, D., . . . Wagner, J. (2014). Laugh When You're Winning. INNOVATIVE AND CREATIVE DEVELOPMENTS IN MULTIMODAL INTERACTION SYSTEMS, 425, 50-+. SPRINGER-VERLAG BERLIN.

Decision letter (RSOS-191815.R0)

17-Jan-2020

Dear Dr Lahnakoski,

The editors assigned to your paper ("Unobtrusive tracking of interpersonal orienting and distance predicts the subjective quality of social interactions") have now received comments from reviewers. We would like you to revise your paper in accordance with the referee and Associate Editor suggestions which can be found below (not including confidential reports to the Editor). Please note this decision does not guarantee eventual acceptance.

Please submit a copy of your revised paper before 09-Feb-2020. Please note that the revision deadline will expire at 00.00am on this date. If we do not hear from you within this time then it will be assumed that the paper has been withdrawn. In exceptional circumstances, extensions may be possible if agreed with the Editorial Office in advance. We do not allow multiple rounds of revision so we urge you to make every effort to fully address all of the comments at this stage. If deemed necessary by the Editors, your manuscript will be sent back to one or more of the original reviewers for assessment. If the original reviewers are not available, we may invite new reviewers.

- Data accessibility

It is a condition of publication that all supporting data are made available either as supplementary information or preferably in a suitable permanent repository. The data accessibility section should state where the article's supporting data can be accessed. This section should also include details, where possible of where to access other relevant research materials such as statistical tools, protocols, software etc can be accessed. If the data have been deposited in an external repository this section should list the database, accession number and link to the DOI

for all data from the article that have been made publicly available. Data sets that have been deposited in an external repository and have a DOI should also be appropriately cited in the manuscript and included in the reference list.

If you wish to submit your supporting data or code to Dryad (<http://datadryad.org/>), or modify your current submission to dryad, please use the following link:
<http://datadryad.org/submit?journalID=RSOS&manu=RSOS-191815>

- **Competing interests**

- **Authors' contributions**

- **Acknowledgements**

- **Funding statement**

on behalf of Dr Antonia Hamilton (Associate Editor) and Essi Viding (Subject Editor)
openscience@royalsociety.org

Comments to Author:

Reviewers' Comments to Author:

Reviewer: 1

Comments to the Author(s)

This paper reports on measures of interpersonal coordination and proxemics from a kinect motion sensor system which they compare to the measures provided by a wearable sensor. I am very sympathetic to this work, as the advent of kinect motion capture technology has the potential to revolutionise several fields, and knowing how accurate or otherwise the data obtained is is clearly necessary before the methodology becomes relied upon as if the gold-standard. However, there were a couple of issues that need to be addressed before this paper is published. Firstly, there were a few key references (addressing the same concerns as the papers authors) that are missing that should be referenced (e.g. Khoshelham and Elberink, 2012; Lavelle et al., 2012; Nichols et al., 2017). Secondly, the structure of the paper was somewhat confusing with the split of information between the methods and results not clearly delimited or signalled so that when first encountering information about, for example, the annotation scheme and classifiers this is incomplete and confusing and it was not clear to me exactly what you had done. Splitting the information between the method and results is fine, as long as the reader is made aware that the expected information is coming (details of the major confusions are provided in the 'specific comments' below). One other major concern I had is that the authors introduced a control condition (p.15 l.28; which is, in my view, a very good thing), but it is never clear exactly how this control condition is generated or whether/how this has been used in the analysis (I could not find any reference to it in the results, and in the discussion it claims it is what figure 5 shows, but this is not what the caption for figure 5 says).

In the results I have some questions about why you chose to use individuals as the unit of analysis for some sections, but both individuals and dyads for other sections. There are also issues about the reporting of significant effects between trials -- these periods are not mentioned in the methods as something you would be analysing which gives the reporting of them the character of a 'fishing expedition' (performing multiple statistical tests until you find a significant result, rather than performing specific tests to address specific hypotheses). These need to either be already part of the planned analysis from the start of the paper, or removed.

The discussion would be clearer if it directly addressed the three hypotheses brought up on page 8 (l.51).

Specific comments.

p.14 l.19 "For those timepoints where the two raters disagreed, a third rater was employed to break the tie." What was the frequency of this? What were raw kappas? (Aha, these are answered in the results: please make a note of this to guide the reader). Did the third annotator know the two tie-break options or choose their annotation based on all 4 options?

p.14 l.35 "Combinations of these data points (yaw, yaw+pitch, yaw+pitch+location, yaw+pitch+location+gaze vector) were used to train Naïve Bayesian classifiers to detect the four gaze categories using random balanced subsamples of the data." This is uninformative. Which did you use? What was the accuracy/recall or f-score? How do I know these were any good at performing this classification task? -- again, some of this comes up in the results (importantly not which classifier you ended up using, which is highly relevant, or the accuracies for other categories), but the reader needs to at least be aware it is coming. I don't actually know what you've done here.

Results: First section seems to be about individuals; have you compared how dyadic partners rated the interaction in relation to each other?

Table 2 (correlation matrix) would be clearer if repeated values were removed (i.e. those above the diagonal)

p.23 l.8 "At the trial level, these effects were only significant during the conversation moments and near significant during pauses between trials, when experimental control was minimal.

Additionally, distance between participants correlated positively with the mean autism quotient of the dyad, but only during the pauses between trials." You've never even stated that you are going to look at pauses between trials, so it reads as if this findings are the results of statistical 'fishing'. Also, you need some numbers (how 'near significant'? $p=0.55$? $p=0.1$?)

p.24 1.5 "per say"  "per se"

References:

Khoshelham, Kourosch & Oude Elberink, Sander. (2012). Accuracy and Resolution of Kinect Depth Data for Indoor Mapping Applications. *Sensors* (Basel, Switzerland). 12. 1437-54. 10.3390/s120201437.

M Lavelle, PGT Healey, R McCabe (2012) Is nonverbal communication disrupted in interactions involving patients with schizophrenia? *Schizophrenia bulletin* 39 (5), 1150-1158

Nichols, Julia K., Sena, Mark P., Hu, Jennifer L., O'Reilly, Oliver M., Feeley, Brian T. and Lotz, Jeffrey C. (2017). A Kinect-based movement assessment system: marker position comparison to Vicon. *Computer Methods in Biomechanics and Biomedical Engineering* 20(12), 1289-1298, 10.1080/10255842.2017.1340464

Reviewer: 2

Comments to the Author(s)

The paper presents a low-cost movement sensor tracking approach to investigate the correlations between quality of social interaction experience and dyad's behaviour. Kinect 2 is used and evaluated against a more robust tracking system. New measures of dyad's behaviour are proposed based on the sensed data. These are complemented by eye-gazing manual annotations of random selected samples. Qualitative self-reports are used to characterise the social experience of the dyad. The results suggest that typical assumptions of interpersonal synchrony may not predict the quality of social interaction and instead patterns of proxemics behaviour may be more informative. The authors also claim that the results show the importance of reducing experimental control to investigate these questions. Data (but not videos) and software are made available to the community.

The paper is interesting and timely as it explores the opportunity that sensing technology offer to really understand people behaviour in relation to experience in real-world more complex settings (or close to). The study is supported by the automatic and manual analysis of the behaviour of 10 dyads in different conditions. The paper is overall well written but the lack of detail makes the understanding of the results and soundness of the analysis difficult.

LITERATURE REVIEW

- The authors should provide a review of the studies on the use of Kinect, RFID or another low cost sensors to study social interaction qualities (e.g., Mancini et al., 2014 & Cattuto et al. 2010) and the features used to understand proxemics behaviour and joint attention using such sensors. They should also look at work in the area of affective computing aimed at detecting emotional responses in dyads using such movement sensors.
- A discussion on the selection of Kinect vs other type of sensors is also welcome as Kinect is quite limited in less static situations. This is particularly important given the accent made on the need for reduced experimental control.

SYSTEM VALIDATION

- Clarify how many participants were used for this validation.

DATA ANALYSIS

Detailed explanation of the data analysis and modelling should be provided. It is unclear what data are used in each of the analysis and if the process is sound.

- T-test: What data were used for the various t-test? Multiple extracts from the same dyads? What are the paired data? The data are clearly not independent and not paired related. Or just mean values over the full sessions? Why just t-test given that there are 3 conditions. Why not a more comprehensive test that take into account the different conditions?

- Classifiers: how are the classifiers trained and tested? Using training samples that are closely related to the testing samples (e.g., close by along the time dimension) lead to higher performances than using more time-unrelated data. Leave one subject out would be better to understand the generalization capabilities of the classifiers. Number of samples per class should be provided.

FINDINGS: the results from the classification are above chance levels but not very high for all classes. Using the classifiers to drive conclusions on what features are more relevant to understand behaviour quality may be risky. E.g., The fact that some gaze behaviour can be predicted by head data alone does not lead to the conclusion that the approach can be used for eye-gazed type prediction. Similarly, it is not possible to conclude on the correlation between classified eye-gaze type and quality of interaction as many eye-gaze types were misclassified. Such claims should be reduced or discussed in relation to the limitation of the approach.

DISCUSSION: it is unclear how the authors concludes that proxemics behaviour is more present when experimental control is minimized. Shouldn't the study consider a variety of contexts? This should be elaborated and grounded in the results. Differences in synchrony between the different conditions should be discussed in relation to the type of movement imposed by the specific tasks. The current analysis does not allow to take into account this factor.

Overall, the work is interesting but a more detailed description of the study and more critical analysis of the contribution made and the limitation of the features proposed should be provided.

Cattuto C, Van den Broeck W, Barrat A, Colizza V, Pinton JF, Vespignani A. Dynamics of person-to-person interactions from distributed RFID sensor networks. *PLoS One*. 2010;5(7):e11596. Published 2010 Jul 15. doi:10.1371/journal.pone.0011596

Mancini, M., Ach, L., Bantegnie, E., Baur, T., Berthouze, N., Datta, D., . . . Wagner, J. (2014). Laugh When You're Winning. *INNOVATIVE AND CREATIVE DEVELOPMENTS IN MULTIMODAL INTERACTION SYSTEMS*, 425, 50-+. SPRINGER-VERLAG BERLIN.

Author's Response to Decision Letter for (RSOS-191815.R0)

See Appendix A.

RSOS-191815.R1 (Revision)

Review form: Reviewer 1

Is the manuscript scientifically sound in its present form?

Yes

Are the interpretations and conclusions justified by the results?

Yes

Is the language acceptable?

Yes

Do you have any ethical concerns with this paper?

No

Have you any concerns about statistical analyses in this paper?

No

Recommendation?

Accept with minor revision (please list in comments)

Comments to the Author(s)

The authors have done, in my view, an excellent job of addressing my concerns from the previous round of reviews. The only comments I have outstanding are extremely minor, and I look forward to seeing the paper published.

I had a question about the looking away category - I think it needs clarifying somewhere that the category includes cases where one person is looking away and the other is looking at them, cases where one person is looking away and the other is looking at the target and cases where both are looking away. You should mention why you have combined these into one group (I'm guessing it's because there aren't enough of these different types of cases, but maybe it would help to have a tabular overview of how much of the time in your annotated sample gaze is in each of the 4 headline categories). This should also help explain why this category is harder to classify, and should at least be mentioned.

p14 line 32: each of the participants was were looking  each of the participants was looking

p15 line 39: dyads making the overall accuracy inflated because of  dyads inflating the overall accuracy due to

p18 Table 1: headings are misaligned

p25 line 6: significant and near significant correlations are reported in the text -- please include the numbers.

p27 line 31: Only significant  The only significant

p28 line 3: Only significant  The only significant

p34 line 50: Additionally, cross-correlation  Additionally, the cross-correlation

p35 line 14: below 95th  below the 95th

p35 line 35: interaction being reflected  interaction would be reflected

p35 line 38: Behavioral synchrony correlates  Behavioral synchrony correlations

p35 line 47: negative findings.  negative findings here.

p37 line 23: forms non-verbal  forms of non-verbal

p37 line 25: on the nature  about the nature

p37 line 44: sensitive for subtle effects  sensitive to subtle effects

Supplementary material:

p2: the Kinect #1  Kinect #1

p2: the Kinect #2  Kinect #2

p2: Correlation are reduced  Correlations are reduced

Supplementary figure 2 caption: violing  violin

p5: face and body orienting  face and body orientation (in subtitle and in text)

p5: but range of body angle was  but the range of body angles was

p5: (displayed in green in). -- MISSING FIGURE REFERENCE

p8: while the second sensors estimates were  while the second sensor's estimates were

Decision letter (RSOS-191815.R1)

Dear Dr Lahnakoski:

On behalf of the Editors, I am pleased to inform you that your Manuscript RSOS-191815.R1 entitled "Unobtrusive tracking of interpersonal orienting and distance predicts the subjective quality of social interactions" has been accepted for publication in Royal Society Open Science subject to minor revision in accordance with the referee suggestions. Please find the referees' comments at the end of this email.

The reviewers and Subject Editor have recommended publication, but also suggest some minor revisions to your manuscript. Therefore, I invite you to respond to the comments and revise your manuscript.

- Ethics statement

- Data accessibility

<http://datadryad.org/submit?journalID=RSOS&manu=RSOS-191815.R1>

- Competing interests

- Authors' contributions

- Acknowledgements

- Funding statement

Because the schedule for publication is very tight, it is a condition of publication that you submit the revised version of your manuscript before 03-Jul-2020. Please note that the revision deadline will expire at 00.00am on this date. If you do not think you will be able to meet this date please let me know immediately.

Supplementary files will be published alongside the paper on the journal website and posted on the online figshare repository (<https://figshare.com>). The heading and legend provided for each supplementary file during the submission process will be used to create the figshare page, so

please ensure these are accurate and informative so that your files can be found in searches. Files on figshare will be made available approximately one week before the accompanying article so that the supplementary material can be attributed a unique DOI.

Reviewer comments to Author:
 Reviewer: 1

Comments to the Author(s)

The authors have done, in my view, an excellent job of addressing my concerns from the previous round of reviews. The only comments I have outstanding are extremely minor, and I look forward to seeing the paper published.

I had a question about the looking away category - I think it needs clarifying somewhere that the category includes cases where one person is looking away and the other is looking at them, cases where one person is looking away and the other is looking at the target and cases where both are looking away. You should mention why you have combined these into one group (I'm guessing it's because there aren't enough of these different types of cases, but maybe it would help to have a tabular overview of how much of the time in your annotated sample gaze is in each of the 4 headline categories). This should also help explain why this category is harder to classify, and should at least be mentioned.

p14 line 32: each of the participants was were looking  each of the participants was looking
 p15 line 39: dyads making the overall accuracy inflated because of  dyads inflating the overall accuracy due to

p18 Table 1: headings are misaligned

p25 line 6: significant and near significant correlations are reported in the text -- please include the numbers.

p27 line 31: Only significant  The only significant

p28 line 3: Only significant  The only significant

p34 line 50: Additionally, cross-correlation  Additionally, the cross-correlation

p35 line 14: below 95th  below the 95th

p35 line 35: interaction being reflected  interaction would be reflected

p35 line 38: Behavioral synchrony correlates  Behavioral synchrony correlations

p35 line 47: negative findings.  negative findings here.

p37 line 23: forms non-verbal  forms of non-verbal

p37 line 25: on the nature  about the nature

p37 line 44: sensitive for subtle effects  sensitive to subtle effects

Supplementary material:

p2: the Kinect #1  Kinect #1

p2: the Kinect #2  Kinect #2

p2: Correlation are reduced  Correlations are reduced

Supplementary figure 2 caption: violating  violin

p5: face and body orienting  face and body orientation (in subtitle and in text)

p5: but range of body angle was  but the range of body angles was

p5: (displayed in green in). -- MISSING FIGURE REFERENCE

p8: while the second sensors estimates were  while the second sensor's estimates were

Author's Response to Decision Letter for (RSOS-191815.R1)

See Appendix B.

Decision letter (RSOS-191815.R2)

Dear Dr Lahnakoski,

It is a pleasure to accept your manuscript entitled "Unobtrusive tracking of interpersonal orienting and distance predicts the subjective quality of social interactions" in its current form for publication in Royal Society Open Science.

Please ensure that you send to the editorial office an editable version of the individual files for each table included in your manuscript. You can send these in a zip folder if more convenient. Failure to provide these files may delay the processing of your proof.

Appendix A

Dear editor,

We appreciate the opportunity to revise our manuscript and the helpful comments by both of the reviewers. We have addressed each of the points made by the reviewers as outlined in the responses below.

The points raised by the reviewers are in normal text and our responses are highlighted in **blue**. Similarly, the changed sections have been highlighted in the manuscript in **blue**. Major changes in the text are also included in the responses to the reviewers in *blue italic* font.

In addition, we have added the requested additional analyses to the software repository. During this process, we also noticed some broken dependencies in the code, so we have taken the opportunity to correct these. These changes have no effect on the original analyses, but allow the code to work on other computing environments in addition to the one where it was developed.

We hope that these revisions sufficiently address the concerns of the reviewers and that you find the manuscript acceptable for publication in Royal Society Open Science.

Yours sincerely,

Juha Lahnakoski, D.Sc. (Tech.)

Comments to Author:

Reviewers' Comments to Author:

Reviewer: 1

Comments to the Author(s)

This paper reports on measures of interpersonal coordination and proxemics from a kinect motion sensor system which they compare to the measures provided by a wearable sensor. I am very sympathetic to this work, as the advent of kinect motion capture technology has the potential to revolutionise several fields, and knowing how accurate or otherwise the data obtained is is clearly necessary before the methodology becomes relied upon as if the gold-standard. However, there were a couple of issues that need to be addressed before this paper is published. Firstly, there were a few key references (addressing the same concerns as the papers authors) that are missing that should be referenced (e.g. Khoshelham and Elberink, 2012; Lavelle et al., 2012; Nichols et al., 2017). Secondly, the structure of the paper was somewhat confusing with the split of information between the methods and results not clearly delimited or signalled so that when first encountering information about, for example, the annotation scheme and classifiers this is incomplete and confusing and it was not clear to me exactly what you had done. Splitting the information between the method and results is fine, as long as the reader is made aware that the expected information is coming (details of the major confusions are provided in the 'specific comments' below). One other major concern I had is that the authors introduced a control condition (p.15 l.28; which is, in my view, a very good thing), but it is never clear exactly how this control condition is generated or whether/how this has been used in the analysis (I could not find any reference to it in the results, and in the discussion it claims it is what figure 5 shows, but this is not what the caption for figure 5 says).

In the results I have some questions about why you chose to use individuals as the unit of analysis for some sections, but both individuals and dyads for other sections. There are also issues about the reporting of significant effects between trials -- these periods are not mentioned in the methods as something you would be analysing which gives the reporting of them the character of a 'fishing expedition' (performing multiple statistical tests until you find a significant result, rather than performing specific tests to address specific hypotheses). These need to either be already part of the planned analysis from the start of the paper, or removed.

The discussion would be clearer if it directly addressed the three hypotheses brought up on page 8 (l.51).

We appreciate the thoughtful and positively presented critique by the reviewer and have revised the manuscript to address these observations as outlined in response to the specific comments below.

In direct response to points raised here, we have included the additional suggested references as well as some additional papers in the introduction and discussion to give a more comprehensive picture of the previous literature (P.5):

"However, in another study comparing Kinect and Vicon systems, a customized setup employing retro-reflective markers instead of standard skeleton tracking was suggested to be necessary for clinically relevant accuracy of kinematic evaluation [29]. Other studies have looked at specifically at the spatial accuracy of Kinect systems. One study evaluating the accuracy of depth measurements with the first-generation Kinect system found an accuracy from a few millimeters up to a few centimeters toward the maximum range of the system compared with a gold standard laser range finder [30]."

P.30 *“While some evidence exists for disrupted gesturing during interactions in psychiatric disorders, such as schizophrenia [41], further work is needed to evaluate what improvements may still be needed to enable the objective, interaction-based phenotyping of patients.”*

To clarify the analysis approaches, we have now added a summary of the comparisons in the beginning of the data analysis section to outline the tests we will be presenting (P.13):

“To compare the conditions, condition-specific mean effects were first calculated for each dyad. Where applicable, the overall effects of the conditions were first evaluated using repeated measures ANOVAs. For the pairwise comparisons between conditions, we employed paired t-tests. Because all data and residuals were not normally distributed, we repeated the analyses with randomly permuted data. Using this surrogate data as a null distribution yielded almost identical results compared to the parametric test. Therefore, in the following we focus only on the parametric test statistic.”

We have also included subheadings in the Data Analysis section in the Methods, which correspond to the subsections of the Results. By making the link between the analysis and results more explicit, we believe this will also aid the readers understanding of the paper by signaling what is to come in the Results section.

Additionally, we have revised the discussion by directly addressing the hypotheses as suggested.

Beginning of “Experimental and subjective correlates of proxemic behavior” section (bottom of P.31):

“This is in line with our general hypotheses that interpersonal behavior is predictive of the quality of interaction and that the type of interaction affects the level to which interaction quality is reflected in the interpersonal measures. This was particularly true when comparing free conversation to the gameplay conditions, where not only behavior was clearly different but also the level of correlation of behavior and subjective evaluations were differentially sensitive to orienting and distancing.”

At the junction of 2nd and 3rd paragraph of “Interpersonal synchrony and similarity of movement profiles” section we state (P.35):

“However, despite subtle changes, our expectation of clear differences between competitive and cooperative conditions was not fulfilled either in the synchrony or the proxemics measures. This suggests that the tasks were indeed well balanced in overt behavior, which was a goal of the task design, but more sensitive measures should be designed to detect differences in the underlying motivation of the participants in the two gameplay conditions.

Despite observing clear behavioral synchrony in all task conditions, our hypothesis that quality of interaction being reflected as increased interpersonal synchrony was largely not supported by the current data.”

Finally, we discuss the lack of hypothesized familiarity effects in the new Limitations and future directions section (P.37):

“Finally, we saw no clear effects of familiarity between the interaction partners when comparing the conversation tasks at the beginning and end of the experiment. This could mean either that participants did not show a clear increase in familiarity during the relatively short experiment or, alternatively, that the methods employed here are not sensitive for subtle effects that may have arisen.

Future studies should induce a stronger sense of familiarity between participants, for example, through repeated social interactions across longer time periods as is the case in therapeutic setting, to directly test these hypotheses."

Specific comments.

p.14 l.19 "For those timepoints where the two raters disagreed, a third rater was employed to break the tie." What was the frequency of this? What were raw kappas? (Aha, these are answered in the results: please make a note of this to guide the reader). Did the third annotator know the two tie-break options or choose their annotation based on all 4 options?

We have clarified in the methods that the frequency of agreement is reported in the results section and now also explain the tiebreaker ratings more clearly. In short, the tiebreaker ratings were used for each binary question separately ("looking at partner" and "looking at target") so each question always had a majority response (P.14):

"...two independent raters annotated, with two binary questions, whether each of the participants was were looking at their partner or a potential target of joint attention [...] For those timepoints where the two raters disagreed (see results for the agreement rate between raters), a third rater was employed to break the tie. The third rater rated each binary question separately (look at the partner/target) and the majority (2/3) vote for each question was selected."

p.14 l.35 "Combinations of these data points (yaw, yaw+pitch, yaw+pitch+location, yaw+pitch+location+gaze vector) were used to train Naïve Bayesian classifiers to detect the four gaze categories using random balanced subsamples of the data." This is uninformative. Which did you use? What was the accuracy/recall or f-score? How do I know these were any good at performing this classification task? -- again, some of this comes up in the results (importantly not which classifier you ended up using, which is highly relevant, or the accuracies for other categories), but the reader needs to at least be aware it is coming. I don't actually know what you've done here.

We agree that it is more informative to specifically refer to the upcoming comparison of results in the methods section as well as refer to the results of the comparison directly in the manuscript, and not only in the supplements, where some of this data was now presented. We have, therefore, revised the manuscript directly after the quoted sentence to state (P. 15):

"We compared the performance of these features in the supplementary results and focus only on the latter, best performing set of features in the results."

And in the results (P.21, Predicting gaze behavior from face orientation and location):

"This set of features showed the best performance compared to reduced sets of features (Supplementary Figure 2) and thus we focus only on this best-performing model in the following."

We also now report the mean accuracies over the gaze classes and the corresponding null accuracies in response to a comment on cross-validation by the second reviewer:

"We tested this with a leave-one-dyad-out cross-validation and observed a mean accuracy of ~57% (null max. 26.9%) across gaze categories and dyads, compared with the ~61% mean accuracy (null max. 29.7%) in the randomly split data."

Results: First section seems to be about individuals; have you compared how dyadic partners rated the interaction in relation to each other?

Indeed, the first table is summarizing the individual characteristics of the participants, while the second table focuses on the dyadic scale. We had also compared the ratings of the partners with each other but did not include these in the manuscript. The consensus between participants was quite high for the subjective ratings of relatedness suggesting that this aspect of subjective interaction success was mutual. Other ratings were not significantly correlated. However, the ratings were generally of similar magnitude in all dyads, so the low variance compared with the imprecise subjective evaluations may explain some of this. We now specify that Table 1 shows individual ratings and also report the correlations of the rating across dyads

(Table 1 and the first two paragraphs on P.17):

"...the means and standard deviations of the individual scores and linear correlations of the interaction partners' ratings across dyads are summarized in Table 1"

"The ratings of Relatedness between interaction partners correlated significantly between dyads ($r=0.61$, $p=0.0077$), suggesting that feeling of relatedness was mutual in the dyads. Other evaluations did not show such agreement because of the relatively low variability of ratings between dyads. Age was also correlated between dyads due to one older dyad ($r=0.57$, $p=0.014$)."

Table 2 (correlation matrix) would be clearer if repeated values were removed (i.e. those above the diagonal)

We agree that repeating the same values is not helpful for reading the table because they increase clutter. We have therefore emptied the upper triangle entries and removed the unused row and column as suggested by the reviewer. We have further clarified in the title that these are average ratings of the two individuals in the dyads adding the words *"dyadic average"* (P. 18).

p.23 l.8 "At the trial level, these effects were only significant during the conversation moments and near significant during pauses between trials, when experimental control was minimal. Additionally, distance between participants correlated positively with the mean autism quotient of the dyad, but only during the pauses between trials." You've never even stated that you are going to look at pauses between trials, so it reads as if this findings are the results of statistical 'fishing'. Also, you need some numbers (how 'near significant'? $p=0.55$? $p=0.1$?)

We agree that the pauses between trials need to be mentioned in the methods section and, since these were not the main time windows of interest and we are sorry for this imprecision on our part. We have now clarified that this observation was exploratory and the findings therefore are tentative, even though they follow the intuitive expectations based on e.g. observations of therapists specializing in disorders of social interaction. We have therefore added explanation of where the pauses between trials occur and clarified that these findings are exploratory rather than confirmatory. (P.11, Procedure)

"Between the different tasks, there was a short pause (mean duration across dyads ranging from 50 seconds to 130 seconds; single dyad durations min 32 s, max 260 s) during which the experimenter gave instructions for the next task and/or reset the bundle of sticks on the table. To include a condition without any explicit task demands, we included data during the pauses, when participants sat upright in a neutral position, as an exploratory analysis of interindividual distance; however, this data was not used for analysis"

of synchrony or orienting data, because during the pauses the participants also looked at and talked with the experimenter."

We now also clearly report the statistics for these observations. (P. 24, Results: Proxemic behavior as a predictor of behavioral ratings and personality scores)

"The positive ratings of interest/enjoyment correlated negatively ($r=0.498$, one-tailed $p=0.0177$) with interpersonal distance over the entire experiment (Figure 4 top). In other words, dyads reporting higher mean enjoyment sat (or leaned) closer to each other. Additionally, correlations with relatedness were near significant ($r=-0.3947$, $p=0.0525$) as might be expected due to high similarity of enjoyment and relatedness ratings (see Table 2). At the trial level, these effects were not significant, but remained near the threshold during the conversation moments ($r_{interest}=0.40$ $p=0.05$, $r_{enjoyment}=0.36$, $p=0.07$) and pauses between trials ($r_{interest}=0.34$ $p=0.08$, $r_{enjoyment}=0.35$, $p=0.08$), when experimental control was minimal. During gameplay these correlations were low.

Additionally, we have expanded the discussion on this point to include some caveats in the current experiment (P.32):

"Similarly, in the exploratory analysis focusing on the unconstrained pauses between trials when participants sat neutrally without an explicit need to interact, autistic traits measured by the AQ were correlated with increased distance suggesting that proxemic patterns associated with autistic traits might be most easily observed when people interact freely with no explicit task. This is also in line with clinical insights and procedures for testing autistic behavior (such as the Autism Diagnostic Observation Schedule; ADOS [42]). However, this observation should be tested more thoroughly in future studies. Thus, these Specifically, the gameplay tasks employed here imposed relatively high task demands to the movement of the participant because the instructions dictated a relatively strict turn-taking behavior. Moreover, successful gameplay also required considerable large-scale movements of the upper body as the participants moved around the bundle trying to select the optimal stick for the particular condition. These behavioral requirements imposed by the task may have masked some of the subtle differences in interpersonal distancing that were observed during the discussions and pauses when the posture was more static. However, while the data suggest that participants returned to their neutral sitting position between the pauses, the actions of the interviewer may have had an effect on their body position making the conclusions on this data tentative."

p.24 l.5 "per say"  "per se"

Corrected

References:

Khoshelham, Kourosh & Oude Elberink, Sander. (2012). Accuracy and Resolution of Kinect Depth Data for Indoor Mapping Applications. *Sensors* (Basel, Switzerland). 12. 1437-54. 10.3390/s120201437.

M Lavelle, PGT Healey, R McCabe (2012) Is nonverbal communication disrupted in interactions involving patients with schizophrenia? *Schizophrenia bulletin* 39 (5), 1150-1158

Nichols, Julia K., Sena, Mark P., Hu, Jennifer L., O'Reilly, Oliver M., Feeley, Brian T. and Lotz, Jeffrey C. (2017). A Kinect-based movement assessment system: marker position comparison to Vicon. *Computer Methods*

Reviewer: 2

Comments to the Author(s)

The paper presents a low-cost movement sensor tracking approach to investigate the correlations between quality of social interaction experience and dyad's behaviour. Kinect 2 is used and evaluated against a more robust tracking system. New measures of dyad's behaviour are proposed based on the sensed data. These are complemented by eye-gazing manual annotations of random selected samples. Qualitative self-reports are used to characterise the social experience of the dyad. The results suggest that typical assumptions of interpersonal synchrony may not predict the quality of social interaction and instead patterns of proxemics behaviour may be more informative. The authors also claim that the results show the importance of reducing experimental control to investigate these questions. Data (but not videos) and software are made available to the community.

The paper is interesting and timely as it explores the opportunity that sensing technology offer to really understand people behaviour in relation to experience in real-world more complex settings (or close to). The study is supported by the automatic and manual analysis of the behaviour of 10 dyads in different conditions. The paper is overall well written but the lack of detail makes the understanding of the results and soundness of the analysis difficult.

We thank the reviewer for the positive evaluation and clear suggestions on how to improve the manuscript. Below, we outline the improvements we have made to rectify the inadequacies highlighted by the reviewer.

LITERATURE REVIEW

- The authors should provide a review of the studies on the use of Kinect, RFID or another low cost sensors to study social interaction qualities (e.g., Mancini et al., 2014 & Cattuto et al. 2010) and the features used to understand proxemics behaviour and joint attention using such sensors. They should also look at work in the area of affective computing aimed at detecting emotional responses in dyads using such movement sensors.

We appreciate the suggestions on earlier studies we had missed and have included the papers in the introduction.

In response to the next point, we now briefly discuss the different sensor types and suggest a combination of methods might be the most fruitful direction in the future (see below).

We also briefly discuss detection of e.g. emotional features from motion tracking data, but we have kept it relatively superficial because we do not explore movement fingerprints of specific emotions in the current study (P.36, Limitations and future directions):

"Bodily motion can also reveal aspects of the emotional state of an individual that are readily detectable by human observers, even from extremely simplified point-light displays of human motion without seeing the shape of the body (e.g. [49,50]). Machine learning methods have also made it possible to use such information to computationally detect the emotional state [51] and laughter [52] of individuals based on their motion tracked by the Kinect, which could be beneficial for deeper understanding of the kind of

behavioral differences reported here. Thus, expanding the analyses here with emotion detection may be an interesting avenue for future research."

- A discussion on the selection of Kinect vs other type of sensors is also welcome as Kinect is quite limited in less static situations. This is particularly important given the accent made on the need for reduced experimental control.

This is an important point; indeed, Kinect-based measurement is not easily applicable to some behaviors that include locomotion in large areas or between multiple locations due to the limited range and field of view of the system. To an extent, this limitation can be alleviated by introducing additional sensors, but mobile tracking applications would be better served by other types of tracking systems, which may have other limitations of their own. We now discuss this limitation in the discussion section.

P.37, Limitations and future directions:

"While the current motion tracking system is useful for tracking natural face-to-face social interactions, such as those between a client and therapist during a diagnostic interview or therapy session, it is limited to relatively small and static spatial locations because of the range and field of view of the tracking system. To address more unconstrained daily activities, wearable technologies can extend measurements to real-life conditions. For example, radio frequency identification tags have been used to measure simple aspects of interactions, such as number and length of interactions in freely interacting individuals at a workplace break room [53] or in a conference setting [54] to evaluate social interaction patterns between individuals. With the advent of smart phones, digital phenotyping [55] has allowed measuring things like movement and digitally mediated social interactions in everyday life with the participants' own personal devices. However, most wearable devices do not allow a detailed characterization of subtle forms non-verbal communication such as bodily and facial gesturing, which may offer important information on the nature of interactions. Therefore, these approaches can reveal complementary information and the most fruitful approaches of future research may be a combination of detailed body tracking during relatively static interactions combined with a day-to-day characterization of daily life at a larger scale using wearable devices or smart phones."

SYSTEM VALIDATION

- Clarify how many participants were used for this validation.

This information has been clarified in the text (bottom of P.18):

"...in all 18 dyads"

DATA ANALYSIS

Detailed explanation of the data analysis and modelling should be provided. It is unclear what data are used in each of the analysis and if the process is sound.

- T-test: What data were used for the various t-test? Multiple extracts from the same dyads? What are the paired data? The data are clearly not independent and not paired related. Or just mean values over the full sessions? Why just t-test given that there are 3 conditions. Why not a more comprehensive test that take into account the different conditions?

Unfortunately, it seems our presentation of the analyses was not clear enough and indeed we could have first presented an omnibus test before going into pairwise comparisons between conditions. Indeed, we

tested the mean values/vectors of each condition within each dyad in a paired manner, that is, each dyad had multiple repeated measurements in different conditions.

We have now clarified in the methods section that we tested the mean results of each condition across dyads and have added the interaction results from repeated measures ANOVAs before introducing the pairwise tests as post-hoc analyses. The ANOVA results, as expected, concur with the pairwise comparisons between conditions. Because the data/residuals did not pass the test for normality, we have additionally repeated the analysis with randomly permuted data to confirm that this violation does not alter our conclusions (P.13):

“To compare the conditions, condition-specific mean effects were first calculated for each dyad. Where applicable, the overall effects of the conditions were first evaluated using repeated measures ANOVAs. For the pairwise comparisons between conditions, we employed paired t-tests. Because some of the data and residuals were not normally distributed, we repeated the analyses with randomly permuted data. Using this surrogate data as a null distribution yielded almost identical results compared to the parametric test. Therefore, in the following we focus only on the parametric test statistics.”

For the joint orienting histograms, the significant regions in the ANOVA were very similar to those of the pairwise comparisons between conditions. We have added the contour of significant main effects of condition in Figure 2 and added the following text (P.20):

“Using repeated measures ANOVAs on the smoothed (Gaussian kernel, $\sigma=5^\circ$) gaze heatmaps, we observed significant main effect of condition in several parts of the joint orienting space, shown in gray dashed lines in Figure 2. Evaluating the underlying effects more thoroughly in paired t-tests between conditions...”

For the durations of predicted gaze behaviors, we have added the following (P.23):

*“Using the prevalence of the predicted gaze behaviors (Figure 3 C), we observed highly significant interaction between task and gaze behavior type ($F= 38.9$, $p=2.8*10^{-35}$). In pairwise comparison of specific gaze behaviors between conditions, we observed...”*

For the windowed cross-correlations of head movement, we have added ANOVA results as bars at the bottom of each plot in Figure 5. As expected, these co-occur with the T-test results.

For cross-correlations of the head, we have added the following text (P. 26):

“Comparing the cross-correlation functions of head speed between conditions, we observed significant ($p<0.05$) main effect of condition in 52.5% of all time delays (dark gray bars in Figure 5). In the pairwise comparisons between conditions...”

For windowed cross-correlations of hand movements, we report (P.27):

“ANOVAs showed significant main effects for condition in 14.95% of time points. Only significant differences in the pairwise comparisons...”

- Classifiers: how are the classifiers trained and tested? Using training samples that are closely related to the testing samples (e.g., close by along the time dimension) lead to higher performances than using more time-unrelated data. Leave one subject out would be better to understand the generalization capabilities of the classifiers. Number of samples per class should be provided.

We have now clarified the cross-validation strategy of the classifiers. In the initial submission, for the evaluation of the classifier performance, we used half of the annotated data for training and half for testing. These were selected randomly from all dyads and all trials and the analysis was repeated 50 times and average accuracies and confusion matrices were created. Indeed, it is also informative to use a leave-one-subject-out cross-validation, which should be more conservative and measure more directly the generalizability of the classifier to new participants. Therefore, we now also report these accuracies for completeness, although this is not the main message of the paper.

In the methods, we have added a more thorough description of the cross-validation (P.15):

“The goal of the classifier was to test whether we can use a small subset (few seconds) of the data to make predictions about the full data (tens of minutes). To evaluate this, we performed 50 cross-validation iterations using half of the combined samples from all participants for training the classifier and the other half for testing. The classifier was then also applied to the full data to predict the overall amount of gaze behaviors in the full data. Additionally, we tested the generalizability of the classifier similarly, but this time using leave-one-dyad-out cross-validation, training on all-but-one dyad and testing on the left out dyad. This process was repeated for each dyad and average performance across dyads and categories was calculated. We report the average class accuracies rather than overall accuracy, because the categories could not be balanced for the individual dyads making the overall accuracy inflated because of high accuracies for the overrepresented categories.”

In the results, we further state (P.24):

“Here, we focused on predicting the amount of each gaze behavior in the full data based on a small manually labeled subsample to see whether the face orientation could be used as a proxy for gaze of the participants. While we observed encouraging results, especially for eye contact, in some applications it would be preferable to be able to predict the gaze in unseen participants. We tested this with a leave-one-dyad-out cross-validation and observed a mean accuracy of ~57% (null max. 26.9%) across gaze categories and dyads, compared with the ~61% mean accuracy (null max. 29.7%) in the randomly split data. While the performance was still well above chance level, variability across dyads was relatively high. While the performance was still well above chance level, variability across dyads was relatively high. This was particularly true for eye contact, which was not observed at all for some participants (Figure 3 C) leading to 0% accuracies for the category and thus reducing the mean across dyads.”

Additionally, we discuss the limitations of the classifier as outlined in response to the next point below.

FINDINGS: the results from the classification are above chance levels but not very high for all classes. Using the classifiers to drive conclusions on what features are more relevant to understand behaviour quality may be risky. E.g., The fact that some gaze behaviour can be predicted by head data alone does not lead to the conclusion that the approach can be used for eye-gazed type prediction. Similarly, it is not possible to conclude on the correlation between classified eye-gaze type and quality of interaction as many eye-gaze types were misclassified. Such claims should be reduced or discussed in relation to the limitation of the approach.

We agree that some eye gaze types are not well predicted and even eye contact and joint attention are not perfect, although they may still provide a useful proxy for a tentative evaluation. We have thus clarified the limitations in the text and toned down the way these results are presented accordingly (P. 33–34):

"Therefore, even if the face direction and location predicted all gaze categories well above chance, only eye contact was detected at a level that strongly predicted the amount of the behavior at an individual level, as suggested by the correlations of real and predicted gaze behavior durations across dyads. Yet, even the accuracy of 70–80% attained for eye contact and joint attention means that the confidence for any single timepoint is not extremely high despite the classifier producing good congruence over the entire experiment. Here, our goal was to test whether we can predict gaze from the Kinect data to see whether labeling only a small subset of data would allow us to infer the gaze behavior during the rest of the interaction. However, if the goal is to use the detection scheme as a predictor of gaze in unseen participants, as might be the case in clinical studies, the generalizability of the model across subjects should be improved from the ~57% mean accuracy achieved here. These imperfect prediction results are attributable not only to the imperfect correspondence of eye and face directions and interindividual differences, but also to the relatively noisy head orientation data. Thus, further work should seek to improve both data quality and prediction accuracy, which may be achieved by future improvements to the tracking hardware itself as well as optimizing the positioning of the sensors in relation to the individuals being tracked and further development of motion tracking and classification algorithms."

DISCUSSION: it is unclear how the authors concludes that proxemics behaviour is more present when experimental control is minimized. Shouldn't the study consider a variety of contexts? This should be elaborated and grounded in the results. Differences in synchrony between the different conditions should be discussed in relation to the type of movement imposed by the specific tasks. The current analysis does not allow to take into account this factor.

It seems that the basis of these interpretations were not explained clearly enough and we are happy to rectify this. Admittedly, these points are still tentative and should not be taken as a ground truth, but rather topics for further examination. We have now extended the discussion on the movement confounds imposed by the tasks and how these types of experimental limitations can confound the types of effects that will be visible in the data.

P. 32, on limitations due to task demands:

"Similarly, in the exploratory analysis focusing on the unconstrained pauses between trials when participants sat neutrally without an explicit need to interact, autistic traits measured by the AQ were correlated with increased distance suggesting that proxemic patterns associated with autistic traits might be most easily observed when people interact freely with no explicit task. This is also in line with clinical insights and procedures for testing autistic behavior (such as the Autism Diagnostic Observation Schedule; ADOS [42]). However, this observation should be tested more thoroughly in future studies. Specifically, the gameplay tasks employed here imposed relatively high task demands to the movement of the participant because the instructions dictated a relatively strict turn-taking behavior. Moreover, successful gameplay also required considerable large-scale movements of the upper body as the participants moved around the bundle trying to select the optimal stick for the particular condition. These behavioral requirements imposed by the task may have masked some of the subtle differences in interpersonal proxemics behavior that were observed during the discussions and pauses when the posture was more static. However, while the data suggest that participants returned to their neutral sitting position between the pauses, the actions of the interviewer may have had an effect on their body position making the conclusions on this data tentative."

Overall, the work is interesting but a more detailed description of the study and more critical analysis of the contribution made and the limitation of the features proposed should be provided.

We appreciate the encouraging comments and hope we have sufficiently addressed the limitations in the our responses above.

Cattuto C, Van den Broeck W, Barrat A, Colizza V, Pinton JF, Vespignani A. Dynamics of person-to-person interactions from distributed RFID sensor networks. PLoS One. 2010;5(7):e11596. Published 2010 Jul 15. doi:10.1371/journal.pone.0011596

Mancini, M., Ach, L., Bantegnie, E., Baur, T., Berthouze, N., Datta, D., . . . Wagner, J. (2014). Laugh When You're Winning. INNOVATIVE AND CREATIVE DEVELOPMENTS IN MULTIMODAL INTERACTION SYSTEMS, 425, 50-+. SPRINGER-VERLAG BERLIN.

Appendix B

We thank the reviewer for the positive evaluation and the clear correction suggestions to the text.

We have applied all corrections to the typographical errors as suggested, and included the p-values for the correlations between distance and age.

In addition, we have added further explanation on the question below. Our response is highlighted in blue and the added text in the manuscript is quoted in the response. Changes in the manuscript are also highlighted in blue.

Q: I had a question about the looking away category - I think it needs clarifying somewhere that the category includes cases where one person is looking away and the other is looking at them, cases where one person is looking away and the other is looking at the target and cases where both are looking away. You should mention why you have combined these into one group (I'm guessing it's because there aren't enough of these different types of cases, but maybe it would help to have a tabular overview of how much of the time in your annotated sample gaze is in each of the 4 headline categories). This should also help explain why this category is harder to classify, and should at least be mentioned.

A: The reviewer is right that this definition leads to some ambiguous cases in the “away” category. The moments when both participants look away at the same time were also under-represented in the data compared to just one person looking away. Initial reason for defining this category was the behavioral significance of moments when a person looks away from the partner. This usually happened during conversation moment, and the definition is ambiguous in some cases as the “other-oriented gaze” category could have been equally fitting. We have included this point in the discussion and now include the prevalence of all combinations of gaze behaviors as a supplementary table:

P.34: “One additional aspect that may have explained the poorer performance in some classes of dyadic behavior is the ambiguity of the gaze categories themselves. In particular, some timepoints in the away category contained moments when only one participant was looking away while their partner was looking at them, which could have equally well been categorized as other-oriented gaze had we defined the categories differently. This definition was motivated by the behavioral significance of either person looking away from the other as well as the low number of samples where both participants were looking away at the same time (~2% of training data; see Supplementary Table 1). However, this may have made these two classes more heterogeneous and ambiguous than the others, thereby making them more difficult to classify.”

Supplementary material P.3:

Supplementary Table 1: Prevalence of all combinations of gaze behaviors across all trials

Partner Partner	Target Target	Partner Target	Away Partner	Away Target	Away Away	Total
11.7%	66.8%	8.6%	7.8%	2.9%	2.1%	100.0%